# Are Common Substructures Transferable? Riemannian Graph Foundation Model with Neural Vector Bundles

Li Sun [1]   Zhenhao Huang [2]   Yiding Wang [1]   Qin Chen [1]   Pietro Liò [3]   Philip S. Yu [4]

## Abstract

Foundation models have sparked a revolution via a pretraining-adaptation paradigm, with recent efforts extending this success to graphs. Unlike other modalities, graphs contain rich structural patterns, yet their structural transferability remains poorly understood. Prior studies consider common substructures in the discrete realm, and we are motivated by a fundamental question: Are common substructures transferable? The underlying theory is largely underexplored. In this work, we shift toward learning transferable structures through the lens of functional behavior. Theoretically, we connect transferable substructures to intrinsic geometry of the representation space. However, characterizing such intrinsic geometry has rarely been touched. Grounded in Riemannian geometry, we develop a graph intrinsic geometry learning framework—**Neural Vector Bundle**, which enables parsing intrinsic geometry with local coordinates. Building on this, we design GAUGE, a pretrainable neural architecture that constructs the vector bundle, flattening geometrically compatible local coordinates, and a new Dirichlet loss, which also measures the transfer effort. We empirically validate its superior expressiveness in challenging tasks including zero-shot link prediction and graph isomorphism.

## 1. Introduction

Foundation models have driven revolutionary progress in natural language (Brown et al., 2020), computer vision (Rad, 2021), and multimodal reasoning (Li et al., 2024). Recently, significant efforts have aimed to replicate such success in the graph, given its ubiquity in real-world applications. The goal is to develop a universal representation mechanism capable of capturing shared structural patterns across diverse graphs and supporting a broad spectrum of downstream tasks.

A key distinguishing characteristic of graphs, compared to other modalities, is their rich structural patterns. While large language model (LLM)-based methods show promise for text-attributed graphs (Zhu et al., 2025; Xia et al., 2024; Chen et al., 2024; 2025; Ren et al., 2024), their text reliance limits their wide usage particularly for the prevalent text-free graphs. Another line of research ties structural transferability to common substructures, such as motifs (Sun et al., 2025b), graphons (Yuan et al., 2025; Wang et al., 2025b), or structural vocabularies (Wang et al., 2025c; Bo et al., 2025). Delving deeper into structural transferability, our work is motivated by a fundamental question: *Are common substructures truly transferable?* In language, a word derives meaning from its local context; analogously, in graphs, a substructure derives its function from in the surrounding structural neighborhood, and a graph learning model ultimately centers on this functional behavior, namely, how it operates on the graph structure. However, a principled framework for understanding such **behavioral invariance** is still lacking: which structural behaviors learned during pretraining transfer to target graphs? The theoretical underpinnings of this question remain underexplored.

In this paper, departing from prior studies that identify common substructures in discrete realm, we shift toward learning transferable structures through the lens of functional behavior. Our solution begins with the intuition that, if a structural behavior learned during pretraining is transferable, it requires little adaptation when applied to target graphs. In other words, its invariant behavior is not affected by the structural neighborhood. Our first contribution is to discover the connection between behavioral invariance and geometric flatness in the representation space. This implies that structures associated with geometrically flat representations are transferable, although identifying such flatness is nontrivial.

Riemannian geometry offers an elegant framework for analyzing graph structure geometrically, yet significant challenges remain. In the literature, a series of Riemannian models have been proposed for graph representation learn-

[1]Beijing University of Posts and Telecommunications, Beijing, China [2]North China Electric Power University, Beijing, China [3]University of Cambridge, Cambridge, UK [4]University of Illinois Chicago, IL, USA. Correspondence to: Li Sun <lsun@bupt.edu.cn>.

*Proceedings of the 43rd International Conference on Machine Learning*, Seoul, South Korea. PMLR 306, 2026. Copyright 2026 by the author(s).

ing (Chami et al., 2019; Bachmann et al., 2020; Xiong et al., 2022), and more recently, for graph foundation models (Sun et al., 2025b). These methods primarily represent graphs in predefined, extrinsic geometric priors (Chen et al., 2022; Gu et al., 2018; Zhang et al., 2021), but they fail to characterize the intrinsic geometry underlying representations generated by usual graph models. Bridging this gap, our second contribution is the development of a new framework for intrinsic graph geometry learning—**Neural Vector Bundle**—designed to systematically analyze transferable structures during pretraining. The novelty lies in introducing a vector bundle formulation to model intrinsic graph geometry. In this framework, the graph structure is preserved in an abstract base manifold, while each local region (i.e., node) is equipped with an attached fiber (i.e., a vector space) depicting its local structure. To enable this, we start from the concept of local trivialization and formalize a neural implementation of local coordinates and parallel transport.

Based on these foundations, we propose a novel **gra**ph **u**niversal pretraining architecture with intrinsic **ge**ometry learning (referred to as GAUGE) for more accurate knowledge transfer and better expressiveness in downstream tasks. Specifically, GAUGE constructs a neural vector bundle by progressively flattening neighboring fibers that are geometrically compatible, and learning invariant substructures through their connection to geometric flatness. Our third contribution is discovering that Dirichlet energy can serve as the measure of behavioral invariance, and we thus formulate a Dirichlet loss to jointly learn the representation and invariant substructures. This loss serves as a measure of transfer effort and specifies the behavior mechanism learned from which substructures is appliable to unseen graphs.

**Contribution Highlights.** Key contributions are three-fold:

- We develop a principled geometric understanding on structural transferability in graph pretraining. Rather than focusing on structures in the discrete realm, we bridge transferable structures to intrinsic graph geometry through the lens of behavior invariance.

- We propose an intrinsic geometry learning framework, Neural Vector Bundle, which characterizes local region of the underlying manifold with parametric local coordinates, grounded in Riemannian geometry.

- We design a novel pretraining architecture GAUGE with a Dirichlet loss that measures transfer effort, and its superior expressiveness is empirically demonstrated on challenging tasks, including graph isomorphism and zero-shot link prediction.

## 2. Related Work

**Towards Graph Foundation Model** The expressiveness of Graph Neural Networks (GNNs) often degrades on new graphs without retraining. Thus, research interest is shifting toward developing pre-trainable, general-purpose neural architectures for universal graph learning. Recently, the success of LLM inspires its extension to text-attributed graphs. However, due to differences between graph and language structures, aligning graphs with LLMs is challenging (Ren et al., 2024). Although GNNs are often incorporated to capture structural patterns, the universality of such hybrid methods primarily comes from LLMs through textual attributes (Chen et al., 2024; Huang et al., 2024). This reliance limits the applicability of LLM-based methods to prevalent text-free graphs (Yu et al., 2025; Zhao et al., 2025c). Meanwhile, there exist efforts toward general-purpose graph learning within narrower domains (Guo et al., 2025a; Liu et al., 2023) or specialized tasks (Zhao et al., 2025a).

**General-Purpose Graph pretraining** Substantial efforts have been devoted to designing universal mechanisms that can learn invariant patterns from diverse graphs. However, identifying such patterns is nontrivial, and existing candidates include motifs (Sun et al., 2025b), trees in message-passing schemes (Wang et al., 2024; 2025c), graphons (Yuan et al., 2025), and various forms of structural vocabularies (Bo et al., 2025; Jiang et al., 2024). On the one hand, prior work often focuses on discrete substructures; in contrast, we explore invariant structural patterns in a continuous geometric space regarding the functional behavior. On the other hand, although existing solutions show promising results, a principled understanding of transferability—which structural pattern learned during pretraining applies to target graphs—remains lacking. Another research direction investigates the adaptation of pre-trained models to downstream tasks via either fine-tuning (Wang et al., 2024; Sun et al., 2026a) or prompt learning (Huang et al., 2025; Wang et al., 2025a; Yu et al., 2025), with recent extensions to continual learning settings (Guo et al., 2025b). We note that this line of work is orthogonal to our focus.

**Riemannian Graph Learning** Existing methods study graph representation learning in a predefined manifold. Research has centered on selecting geometric priors that align with certain graphs (Sun et al., 2026c). For example, hyperbolic spaces are well-suited for hierarchical structures (Chami et al., 2019; Chen et al., 2022; Sun et al., 2026b), while hyperspherical spaces are well-suited for cyclical ones (Liu et al., 2022). Representation spaces can be constructed via products or quotients (Sun et al., 2024b; Zhang et al., 2021; Gu et al., 2018; Xiong et al., 2022; Law, 2021; Sun et al., 2024a). Also, we note that the concepts of sheaves, bundles (Zaghen et al., 2024; Hansen & Geb-

hart, 2020; Bamberger et al., 2025) and Riemannian dynamics (Sun et al., 2025a) have been introduced for designing GNNs, but they have not yet been connected to transferability. In contrast, we focus on the intrinsic geometry of graphs, i.e., the geometry of the underlying data manifold induced by graph representations. This is challenging because the manifold is unknown, and its tools for characterizing it remain underdeveloped.

## 3. A Neural Vector Bundle Framework

Currently, a principled understanding of transferability remains lacking, which limits precise knowledge transfer and the expressiveness of pretrained models. In this paper, we aim to bridge this gap. Notably, we establish a theoretical connection between transferability and the intrinsic geometry of graphs; however, characterizing such geometry has received little attention, posing a significant challenge. To address this, we introduce the concept of vector bundles into graph learning and propose a novel Riemannian deep learning framework—Neural Vector Bundle—that enables a principled geometric understanding for modeling structural transferability. Key notations are summarized in Appendix A, and detailed proofs are provided in Appendix C.

### 3.1. Background

We briefly review the concepts in Riemannian geometry: manifold, Riemannian metric, connection and vector bundle. A graph $\mathcal{G} = (\mathcal{V}, \mathcal{E}, \mathbf{X})$ is defined on node set $\mathcal{V}$ and edge set $\mathcal{E}$ with $\mathbf{X} \in \mathbb{R}^{N \times D}$ summarizing node attributes. In Riemannian geometry, manifold is the continuous counterpart of graph. Specifically, a Riemannian manifold $(\mathcal{M}, \mathrm{g})$ is defined on a $C^\infty$ smooth manifold $\mathcal{M}$ with a $C^\infty$ Riemannian metric g, which assigns each point $\boldsymbol{p} \in \mathcal{M}$ an inner product on its tangent space. A connection $\nabla$ on $\mathcal{M}$ (e.g., Levi-Civita connection) defines how to differentiate vector fields along curves, enabling the notion of parallel transport.

**Vector Bundle**    As a key concept in this paper, it can be roughly understood as a "background" space surrounding the manifold. Specifically, a $C^\infty$ vector bundle with rank $r$ is a $C^\infty$ surjection $\pi : E \to \mathcal{M}$ with base manifold $\mathcal{M}$ and total space $E$ satisfies two conditions: (1) Each point $\boldsymbol{p} \in \mathcal{M}$ is attached to a **fiber** $E_{\boldsymbol{p}} = \pi^{-1}(\boldsymbol{p})$, a real vector space with dimension $r$. (2) There exists an open neighborhood $U \in \mathcal{M}$ such that there is a diffeomorphism

$$\phi_U : \pi^{-1}(U) \to U \times \mathbb{R}^r, \tag{1}$$

which is restricted to a linear isomorphism $E_{\boldsymbol{p}} \to \{\boldsymbol{p}\} \times \mathbb{R}^r$ on each fiber, such that for every $\boldsymbol{q} \in U$, the restriction $\phi_U|_{E_{\boldsymbol{q}}} : E_{\boldsymbol{q}} \to \{\boldsymbol{q}\} \times \mathbb{R}^r$ is a linear isomorphism. Rigorous elaborations are given in Appendix B.

### 3.2. Connecting Transferability to Geometry

Essentially, a graph learning model $\Phi : \mathcal{V} \to \mathbb{R}^d$ defines a real function over $\mathcal{V}$ that generates representations $\boldsymbol{z} \in \mathbb{R}^d$ under structural interactions among the nodes. Thus, we explore the transferability through the lens of functional behavior, departing from previous methods focusing on the discrete realm. Specifically, we first elaborate on the behavioral invariance and further subsequently it to the intrinsic flat geometry in the representation space.

**Transferability & Behavioral Invariance**    Here, we begin with the intuition on knowledge transfer: if a structural behavior learned during pretraining is transferable, it requires little to no adaptation when applied to unseen graphs. In other words, transferability is rooted in behavioral invariance. In a graph $\mathcal{G}$, we consider a substructure $\mathcal{T}$ (a connected component of $\mathcal{G}$) in the representation space. Nodes in $\mathcal{T}$ are unaffected by the nodes outside this substructure or, in other words, $\mathcal{T}$ is invariant to its structural neighborhood.

**Vector Bundle Representation**    In this paper, we consider the **intrinsic geometry**[1]—the geometry of the underlying data manifold from given node representations in $\mathbb{R}^d$, generated by any graph learning model $\Phi$. To this end, we introduce a vector bundle as the representation space. Its abstract base manifold preserves the graph structure, while the attached fibers characterize the local geometry of the manifold—without requiring any geometric prior. In this framework, each node $v_i$ is endowed with both a global representation $\boldsymbol{z}_i$ and a local representation $\boldsymbol{x}_i$ residing in its corresponding fiber. Our theoretical analysis is constructed within this representation framework.

**Connection to Intrinsic Geometry**    We show that the (behavioral) invariant substructure is related to the flatness of its intrinsic geometry. Under vector bundle representation, the invariant substructure $\mathcal{T}$ can be formalized with parallel transport $\boldsymbol{g}$: $\mathcal{T}$ is said to be invariant if $\left\| \boldsymbol{x}_i - \boldsymbol{g}_{ij}(\boldsymbol{x}_j) \right\| \leq \varepsilon$ holds for any $\varepsilon \to 0$ and $v_j \in \mathcal{N}_i$. We analyze the geometry of $\mathcal{T}$ using the concept of holonomy in Riemannian geometry, describing how a vector changes when traversing along a closed curve. In a graph, the holonomy can be composed by parallel transport $\boldsymbol{g}_{ij}$ along a loop.

**Theorem 3.1** (**Invariance & Intrinsic Flatness (Berwick-Evans et al., 2023), Appendix C.1**). *A discrete connection of $E$ is flat if and only if holonomy around every 2-simplex is the identity. If there is no 2-simplex, the flat condition is automatically satisfied.*

It claims that the intrinsic geometry of vector bundle $E$ over an invariant $\mathcal{T}$ is flat. For any 2-simplex $(i, j, k)$, we have

---

[1]The term of intrinsic geometry in this paper is aligned with its meaning in differential geometry (do Carmo, 1992).

$g_{ik}g_{kj}g_{ji} = \mathrm{id}_{E_{v_i}}$ where id denotes the identity mapping.

## 3.3. Intrinsic Graph Geometry Learning

While the vector bundle offers a mathematical framework for studying manifold geometry, the elements in each fiber are abstract vectors. To implement this theory in practice, we propose a data-driven deep learning framework for constructing vector bundles, namely, Neural Vector Bundle. Its key components include local trivialization for the fibers and pseudo parallel transport.

**Local Coordinates**   First, we parse the local coordinate of each fiber in computational valid $\mathbb{R}^r$ from the global representation in $\mathbb{R}^d$. The key is **local trivialization**. Concretely, for each node $v_i$, the local trivialization $\phi_i$ restricted to its fiber $E_i$ defines a linear isomorphism $\phi_i : E_{v_i} \to \{v_i\} \times \mathbb{R}^r$, mapping to $\mathbb{R}^r$ space. Thus, given a graph encoder $\Phi$, we instantiate a parametric local trivialization as $\phi_i = \psi_i \circ \Phi$, where $\psi_i$ is represented by a matrix $\left(\mathbf{Q}^{(i)}\right)^\top \in \mathbb{R}^{r \times d}$ with $(\mathbf{Q}^{(i)})^\top \mathbf{Q}^{(i)} = \mathbf{I}$. We denote the collection of trivializations over open neighborhoods of all nodes by $\mathcal{Q}$. For the subset of $\mathcal{Q}$ over $\mathcal{N}_i$, we denote $\mathcal{Q}^{(i)} = \left[\mathrm{Vec}(\mathbf{Q}^{(j)})\right]_{j \in \mathcal{N}_i}^\top$.

Now we introduce a neural implementation of $\mathcal{Q}$. Our strategy is to first obtain a set of vectors spanning the fiber and then determine an orthonormal basis, giving the matrix form of local trivialization. First, we construct disjoint subspaces to capture the diverse ways in which nodes interact with their neighborhoods. Without loss of generality, for the input matrix of layer $l$, we decompose $\mathbf{Z}^{l-1} \in \mathbb{R}^{N \times d}$ into $r$ disjoint subspaces via learnable projections, yielding $\{\boldsymbol{z}_i^{(h),l-1}\}_{h=1}^r$. Next, we adopt the multi-head attention, and the fiber vector in each head $h$ is derived as follows,

$$\hat{\boldsymbol{q}}_i^{(h),l} = \boldsymbol{z}_i^{l-1} - \frac{1}{\sum_{j \in \mathcal{N}_i} \alpha_{ij}^{(h),l}} \sum_{j \in \mathcal{N}_i} \alpha_{ij}^{(h),l} \boldsymbol{z}_j^{(h),l-1} \quad (2)$$

$$\alpha_{ij}^{(h),l} = \frac{\exp(f^{(h)}(\boldsymbol{z}_i^{(h),l-1} \| \boldsymbol{z}_j^{(h),l-1})/\tau)}{\sum_{k=1}^r \exp(f^{(h)}(\boldsymbol{z}_i^{(h),l-1} \| \boldsymbol{z}_j^{(h),l-1})/\tau)}, \quad (3)$$

where $f^{(h)} : \mathbb{R}^{2d} \to \mathbb{R}$ is a nonlinear learnable function, and $\tau$ controls attention sharpness. Finally, at each fiber $E_{v_i}$, we assemble the $H$ residual vectors into a matrix and apply QR decomposition to obtain an orthonormal basis:

$$\mathbf{Q}^{(i),l} = \mathrm{QR}\left(\left[\hat{\boldsymbol{q}}_i^{(1),l}, \cdot, \hat{\boldsymbol{q}}_i^{(r),l}\right]\right). \quad (4)$$

This yields a local map $\mathbf{Q}^{(i),l} : E_{v_i} \cong_{\phi_i} \mathbb{R}^r \to \mathbb{R}^d$, whose transpose $(\mathbf{Q}^{(i),l})^\top$ is a matrix form of the local trivialization over $\mathcal{N}_i$.

**Parallel Transport**   Then, we specify the map between different fibers. The canonical map in Riemannian geometry is

called parallel transport, which is a vector field $V(t)$ along a smooth curve $\gamma(t) : [a, b] \to \mathcal{M}$ satisfying $\nabla_{\dot{\gamma}(t)} V(t) = 0$ under a certain connection $\nabla$. In our context, a real vector bundle over $\mathcal{G}$ equips each edge $(i, j) \in \mathcal{E}$ a linear isomorphism $g_{ji} = \phi_j \circ \phi_i^{-1} : E_{v_i} \to E_{v_j}$, endowing a discrete connection of the total space $E$ and parallel transport. Thus, we obtain the parallel transport between fibers along edges.

**Definition 3.2** (**Pseudo Parallel Transport along Edge**). Given an edge $(v_i, v_j) \in \mathcal{E}$ and the fibers of its two endpoints $E_{v_i}$ and $E_{v_j}$, the pseudo parallel transport $\mathbf{P}^{(i,j)} : E_{v_j} \to E_{v_i}$ is defined as

$$\mathbf{P}^{(i,j)} = (\mathbf{Q}^{(i)})^\top \mathbf{Q}^{(j)}. \quad (5)$$

Accordingly, $\mathbf{P}^{(i,j)}$ captures the geometric torsion when a vector in fiber $E_{v_i}$ moves to $E_{v_j}$ along an edge.

## 4. GAUGE: Improving Graph Pretraining with Intrinsic Geometry

We propose a pretrainable GAUGE, sketched in Fig. 1. The novelty lies in its intrinsic geometry learning together with node representation. In brief, we progressively construct the neural vector bundle by flattening geometrically compatible fibers, as in Fig. 1A, and simultaneously identify invariant substructures via minimizing the proposed Dirichlet loss, as in Fig. 1B, which is proven to measure behavioral invariance. During adaptation as in Fig. 1C, the pretrained parameters are kept frozen, and only the two linear layers for input encoding and task adaptation are finetuned.

### 4.1. Neural Architecture

In each layer of GAUGE, node representations $\mathbf{Z}$ are updated in account of the intrinsic geometry. Concurrently, a gated aggregation is designed to flatten the vector bundle by aligning geometrically compatible fibers, characterized by their local coordinates $\mathbf{Q}$. The update rules for $\mathbf{Q}$ and $\mathbf{Z}$ are detailed below.

**Updating Q**   Aiming at neural vector bundle construction, we propose to merge the fibers' local coordinates based on their geometric compatibility. To achieve this, we introduce a gated flattening mechanism that selectively aggregates neighboring fibers according to local coordinates. By Definition 3.2, the compatibility between neighboring fibers is measured by the pseudo parallel transport $\mathbf{P}$. Specifically, for each edge $(i, j)$, the gating mechanism is formulated with $\mathbf{P}^{(i,j)}$ as follows,

$$g_{ij} = \frac{\exp(-\mathrm{Tr}(\mathbf{I}_r - \mathbf{P}^{(i,j)})/\tau)}{\sum_{k \in \mathcal{N}_i} \exp(-\mathrm{Tr}(\mathbf{I}_r - \mathbf{P}^{(i,k)})/\tau)} \in (0, 1), \quad (6)$$

where $\beta > 0$ is a sensitive scaler, and $b \in \mathbb{R}$ is a bias. Neighboring fibers with high compatibility yields large $g_{ij}$,

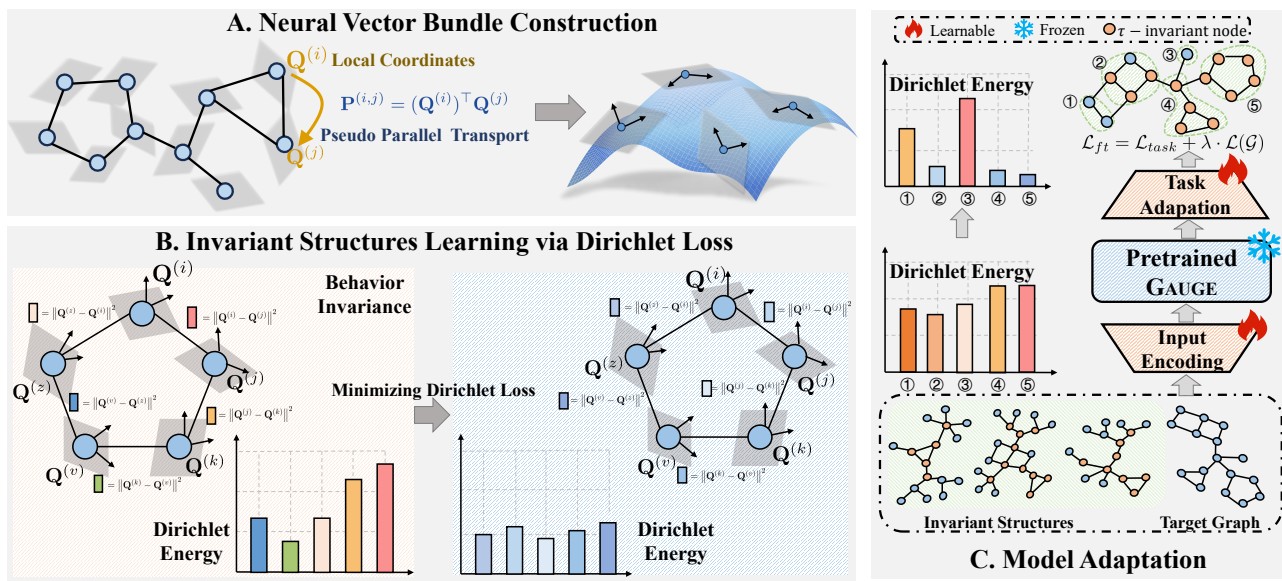

*Figure 1.* Overall Framework. **GAUGE** learns the intrinsic graph geometry while generating node representations, improving model expressiveness and knowledge transfer with invariant substructures.

so that they are selected for aggregations. Thus the local coordinates are updated as follows:

$$\hat{\mathbf{Q}}^{(i),k} = (1-\gamma)\mathbf{Q}^{(i),k-1} + \gamma \sum_{j \in \mathcal{N}_i} g_{ij}\mathbf{Q}^{(j),k-1}, \quad (7)$$

$$\mathbf{Q}^{(i),k} = \mathrm{QR}\left(\hat{\mathbf{Q}}^{(i),k}\right), \quad (8)$$

where $\gamma \in (0,1)$ is the momentum coefficient. The updating rule of $\mathbf{Q}$ iteratively refines local coordinates by smoothing geometrically consistent neighborhoods.

**Updating Z**   Different from the conventional message passing that directly updates node representation in $\mathbb{R}^d$ space, we consider the vector bundle for updating $\mathbf{Z}$ where we leverage the local trivialization for recovering its fiber $E_{v_i}$ and filter noise with the orthogonal complement of its fiber $E_{v_i}^{\perp}$. Additionally, we apply residual connections to allow for the network to go deeper. Thus the global node representation is updated as follows,

$$\tilde{z}_j^{l-1} = \mathbf{Q}^{(j),l-1}\left(\mathbf{Q}^{(j),l-1}\right)^{\top} z_j^{l-1}, \quad (9)$$

$$z_i^l = \frac{1}{|\mathcal{N}_i|} \sum_{j \in \mathcal{N}_i} \mathbf{W}^l \tilde{z}_j^{l-1} + \psi^l(z_i^{l-1}), \quad (10)$$

where $\mathbf{W}^l \in \mathbb{R}^{d \times d}$ is a learnable matrix, and $\psi^l$ is a learnable nonlinear function.

### 4.2. Dirichlet Energy and Transferablity

As elaborated in Sec. 3.2, the invariance is a key to transferablity, yet measuring invariance is challenging. In this

paper, we introduce the concept of Dirichlet energy over vector bundle (Bamberger et al., 2025). Specifically, given a $\mathcal{G} = (\mathcal{V}, \mathcal{E})$ equipped with a vector bundle structure $E$, it is defined as $\mathcal{E}_{\mathrm{Dir}}[E] = \frac{1}{|\mathcal{E}|}\sum_{(i,j)\in\mathcal{E}}\left\|\boldsymbol{x}_i - \mathbf{P}^{(i,j)}\boldsymbol{x}_j\right\|^2$, where $\mathbf{P}^{(i,j)}$ denotes the pseudo parallel transport map from node $v_j$ to node $v_i$ along edge $(i,j)$. We show that the Dirichlet energy gives an upper bound of invariance in terms of local trivialization, where local representation of a node can be inferred from others in the invariant substructure. That is, minimizing Dirichlet energy is sufficient to learn behavioral invariant substructures.

**Theorem 4.1** (**Local Invariance Measure, Appendix C.2.**). *Denote local trivializations at each node $v_i$ as $\mathcal{Q}(v_i) = \mathbf{Q}^{(i)}$. Given node representation $\mathbf{Z} = \{z_i\}_{i\in\mathcal{V}}$ satisfying $\max_i \|z_i\| \leq M$, the following **upper bound** holds,*

$$\mathcal{E}_{\mathrm{Dir}}[E] \leq (4M^2+1)(\mathcal{E}_{\mathrm{Dir}}[\mathcal{Q}] + \mathcal{E}_{\mathrm{Dir}}[\mathbf{Z}]), \quad (11)$$

*where $\mathcal{E}_{\mathrm{Dir}}[\mathcal{Q}]$ and $\mathcal{E}_{\mathrm{Dir}}[\mathbf{Z}]$ denote the Dirichlet energies associated with the local trivializations and the node embeddings, respectively.*

**Theorem 4.2** (**Monotonous Flattening, Appendix C.3.**). *Given a graph $\mathcal{G} = (\mathcal{V}, \mathcal{E})$ and the set of local trivializations $\mathcal{Q}^k$ over $\mathcal{G}$ in the $k$-th layer of gated flattening with $\gamma \in (0,1)$, the layer-wise Dirichlet energy is nonincreasing, $\mathcal{E}_{\mathrm{Dir}}[\mathcal{Q}^k] \leq \mathcal{E}_{\mathrm{Dir}}[\mathcal{Q}^{k-1}]$, and the final local trivializations satisfies the following inequality,*

$$\mathcal{E}_{\mathrm{Dir}}[\mathcal{Q}^k] \leq (1 - C \cdot \gamma)^K \mathcal{E}_{\mathrm{Dir}}[\mathcal{Q}^0], \quad (12)$$

*where $C = 1 - \lambda^2$, and $\lambda$ is the maximal singular value of matrix $\mathbf{G} = [g_{ij}]$.*

---

**Algorithm 1** Training Procedure of GAUGE.

---

**Require:** Graph $\mathcal{G} = (\mathcal{V}, \mathcal{E})$, initial embeddings $\{z_i^{(0)}\}$, model parameters $\Theta$.
 1: **for** each training step **do**
 2:     **for** each network layer $l = 1$ to $L$ **do**
 3:         Compute local coordinates $\{\mathbf{Q}^{(i),l}\}$ via Eqs. 2-4.
 4:         Refine neural vector bundle via a 2-layer gated flattening via Eqs. 6-8.
 5:         Compute node representations $\{z_i^l\}$ via Eq. 10.
 6:     **end for**
 7:     Stop gradient of $\{z_i^{(0)}\}$ by $\hat{z}_i^{(0)} = \text{StopGrad}(z_i^{(0)})$.
 8:     Compute loss $\mathcal{L}(\mathcal{G})$ in Eq. (13).
 9:     Update $\Theta$ via backpropagation.
10: **end for**

---

It claims that GAUGE monotonously lowers the Dirichlet energy over the vector bundle. In other words, it progressively flattens neighboring fibers that are geometrically compatible, and learns invariant substructures through their connection to geometric flatness (Theorem 3.1).

### 4.3. A New Pretraining Loss: Dirichlet Loss

In GAUGE, we present a new pretraining loss, Dirichlet loss, accompanied by several theoretical guarantees. The Dirichlet loss is designed to reveal the intrinsic graph geometry. In addition to representation learning, Dirichlet loss simultaneously characterizes the neural vector bundle, capturing behavioral invariance for more accurate knowledge transfer. For each graph, the Dirichlet loss is formulated in a predictive fashion as follows,

$$\mathcal{L}(\mathcal{G}) = \sum_{i=1}^{N} \left\| \left(\mathbf{Q}^{(i)}\right)^{\top} \hat{z}_i^{(0)} - \frac{1}{|\mathcal{N}_i|} \sum_{j \in \mathcal{N}_i} \left(\mathbf{Q}^{(j)}\right)^{\top} z_j^L \right\|_2^2, \quad (13)$$

where $\hat{z}_i^{(0)} = \text{StopGrad}\left(z_i^{(0)}\right)$. The overall training procedure is summarized in Algorithm 1. This loss will converge to a constant, and vanish over the region—invariant substructures—where the representation of each node can be inferred from its neighbors.

**Recovering Invariant Substructures** As a byproduct, we obtain a measure describing how a substructure differs from behavior invariance, implying the transfer effort during model adaptation to downstream task.

**Definition 4.3** ($\tau$-**Invariance**). Given a graph $\mathcal{G} = (\mathcal{V}, \mathcal{E})$ with node representation $\mathbf{Z} \in \mathbb{R}^{|\mathcal{V}| \times d}$ generated by an encoder $\Phi$ and a connected substructure $\mathcal{T}$, for each node $v_i \in \mathcal{T}$, its representation predictive error is derived as $e(v_i) = \left\| \left(\mathbf{Q}^{(i)}\right)^{\top} (z_i) - \frac{1}{|\mathcal{N}_i|} \sum_{j \in \mathcal{N}_i} \left(\mathbf{Q}^{(j)}\right)^{\top} (z_j) \right\|_2$. For a threshold $\tau > 0$, we say the substructure $\mathcal{T}$ is

$\tau$-invariance under $\Phi$, or $\mathcal{T}$ is $\tau$-invariant if $\chi(\mathcal{T}) = \frac{1}{|\mathcal{T}|} \sum_{v \in \mathcal{T}} e(v) \leq \tau$.

Accordingly, we can recover the invariant substructures, specifying the behavior mechanism learned from which substructures during pretraining are applicable to unseen graphs. The details are provided in Appendix D.

**Geometry of $\tau$-Invariance** The definition above identifies substructures where local representations are mutually predictable. Remarkably, it is sufficient to imply geometric flatness—quasi-identity parallel transport—within this substructure, as formalized below.

**Theorem 4.4** (**Flatness Induced by $\tau$-Invariance, Appendix C.4**). *Let $\mathcal{G} = (\mathcal{V}, \mathcal{E})$ be a graph, and node embeddings are $\mathbf{Z} = \{z_i\}_{i \in \mathcal{V}}$. Suppose $\mathcal{T} \subseteq \mathcal{V}$ is a $\tau$-invariant connected subgraph. Assume that there exists $\delta > 0$ such that $\|\mathbf{z}_i - \mathbf{z}_j\|_2 \leq \delta$ for all $(i, j) \in \mathcal{E}(\mathcal{T})$, and that $\|\mathbf{z}_i\|_2 \geq c > 0$. Then, for every edge $(i, j) \in \mathcal{E}(\mathcal{T})$, the pseudo parallel transport map satisfies*

$$\left\| \mathbf{P}_{(i,j)} - \mathbf{I}_r \right\|_F = \left\| (\mathbf{Q}^{(i)})^{\top} \mathbf{Q}^{(j)} - \mathbf{I}_r \right\|_F \leq C(\tau + \delta),$$

*where $C > 0$ is a constant depending only on $c$ and $r$.*

In particular, as $\tau, \delta \to 0$, the bundle structure over $\mathcal{T}$ becomes geometrically flat.

**Connection to Graph Contrastive Loss** We show that a special case of the proposed Dirichlet loss well approximates graph contrastive learning loss, such as the popular InfoNCE (van den Oord et al., 2018). Specifically, when local coordinates are aligned ($\mathbf{P}^{(i,j)} \approx \mathbf{I}$), minimizing Dirichlet loss pulls together geometrically compatible neighbors—mimicking the "positive" alignment in contrastive learning—while implicitly pushing apart incompatible pairs through high reconstruction error, without requiring explicit data augmentation. It claims that Dirichlet loss can be regarded as a geometric generalization of contrastive loss in light of intrinsic geometry.

**On Structural Transferability** We discuss the transferability during fine-tuning when we adapt the pretrained model to downstream tasks. Following the convention (Sun et al., 2023; Wang et al., 2024), the total fine-tuning loss takes the following form,

$$\mathcal{L}_{\text{ft}} = \mathcal{L}_{\text{task}} + \lambda \cdot \mathcal{L}(\mathcal{G}), \quad (14)$$

where $\lambda > 0$ is a hyperparameter, and $\mathcal{L}_{\text{task}}$ denotes the task-specific loss. In this case, GAUGE performs the downstream task with the pretrained neural vector bundle, preserving behaviorally invariant substructures. We have the following theorem demonstrating that the deviation over invariant substructures is constrained.

*Table 1.* **Main Results on Cross-Domain Transfer.** We report **Mean Accuracy (%)** $\pm$ std over 10 independent runs. Models are pre-trained on source datasets and fine-tuned on target graphs with $k \in \{1, 5\}$ labeled samples (Few-Shot)

| Model | PubMed | | Facebook | | Roman-empire | | Photo | |
|---|---|---|---|---|---|---|---|---|
| | 1-shot | 5-shot | 1-shot | 5-shot | 1-shot | 5-shot | 1-shot | 5-shot |
| GCN (Kipf & Welling, 2017) | $40.91 \pm {\scriptstyle 6.71}$ | $51.34 \pm {\scriptstyle 6.46}$ | $44.43 \pm {\scriptstyle 7.22}$ | $55.25 \pm {\scriptstyle 4.41}$ | $11.29 \pm {\scriptstyle 2.07}$ | $18.19 \pm {\scriptstyle 1.38}$ | $43.03 \pm {\scriptstyle 8.10}$ | $58.34 \pm {\scriptstyle 2.20}$ |
| GraphSAGE (Hamilton et al., 2017) | $41.36 \pm {\scriptstyle 9.78}$ | $51.76 \pm {\scriptstyle 6.83}$ | $47.23 \pm {\scriptstyle 9.05}$ | $\underline{59.93} \pm {\scriptstyle 4.27}$ | $11.34 \pm {\scriptstyle 4.32}$ | $18.24 \pm {\scriptstyle 2.50}$ | $42.53 \pm {\scriptstyle 6.67}$ | $54.55 \pm {\scriptstyle 3.26}$ |
| GAT (Veličković et al., 2018) | $39.86 \pm {\scriptstyle 10.65}$ | $50.37 \pm {\scriptstyle 6.30}$ | $46.26 \pm {\scriptstyle 7.19}$ | $59.82 \pm {\scriptstyle 5.56}$ | $11.71 \pm {\scriptstyle 2.47}$ | $18.20 \pm {\scriptstyle 1.51}$ | $41.74 \pm {\scriptstyle 9.16}$ | $56.96 \pm {\scriptstyle 2.88}$ |
| DGI (Veličković et al., 2019) | $41.64 \pm {\scriptstyle 7.72}$ | $58.20 \pm {\scriptstyle 2.71}$ | $42.57 \pm {\scriptstyle 1.81}$ | $55.66 \pm {\scriptstyle 2.74}$ | $9.32 \pm {\scriptstyle 1.99}$ | $13.74 \pm {\scriptstyle 1.17}$ | $42.36 \pm {\scriptstyle 4.78}$ | $63.20 \pm {\scriptstyle 2.68}$ |
| GraphMAE (Hou et al., 2022) | $43.63 \pm {\scriptstyle 5.81}$ | $56.54 \pm {\scriptstyle 4.85}$ | $43.35 \pm {\scriptstyle 4.98}$ | $58.93 \pm {\scriptstyle 4.60}$ | $16.57 \pm {\scriptstyle 3.23}$ | $17.00 \pm {\scriptstyle 1.35}$ | $44.32 \pm {\scriptstyle 9.74}$ | $62.97 \pm {\scriptstyle 4.38}$ |
| GRACE (Zhu et al., 2020) | $41.04 \pm {\scriptstyle 6.51}$ | $59.66 \pm {\scriptstyle 6.49}$ | $\underline{49.79} \pm {\scriptstyle 7.82}$ | $56.26 \pm {\scriptstyle 5.77}$ | $7.71 \pm {\scriptstyle 3.66}$ | $13.84 \pm {\scriptstyle 5.53}$ | $47.26 \pm {\scriptstyle 6.00}$ | $66.06 \pm {\scriptstyle 4.33}$ |
| GFT (Wang et al., 2024) | $52.39 \pm {\scriptstyle 9.07}$ | $63.11 \pm {\scriptstyle 4.72}$ | $45.79 \pm {\scriptstyle 3.02}$ | $46.33 \pm {\scriptstyle 2.36}$ | $18.63 \pm {\scriptstyle 4.80}$ | $\underline{26.60} \pm {\scriptstyle 2.75}$ | $51.83 \pm {\scriptstyle 5.75}$ | $76.53 \pm {\scriptstyle 2.96}$ |
| RAGraph (Jiang et al., 2024) | $51.39 \pm {\scriptstyle 6.93}$ | $65.71 \pm {\scriptstyle 3.45}$ | $46.01 \pm {\scriptstyle 0.93}$ | $48.88 \pm {\scriptstyle 1.54}$ | $10.51 \pm {\scriptstyle 2.33}$ | $20.99 \pm {\scriptstyle 2.17}$ | $50.62 \pm {\scriptstyle 11.25}$ | $70.28 \pm {\scriptstyle 6.20}$ |
| SAMGPT (Yu et al., 2025) | $43.51 \pm {\scriptstyle 8.43}$ | $68.81 \pm {\scriptstyle 8.98}$ | $47.88 \pm {\scriptstyle 10.23}$ | $56.38 \pm {\scriptstyle 7.69}$ | $10.39 \pm {\scriptstyle 6.15}$ | $25.76 \pm {\scriptstyle 6.20}$ | $49.03 \pm {\scriptstyle 10.87}$ | $71.52 \pm {\scriptstyle 7.26}$ |
| GCOPE (Zhao et al., 2024) | $44.78 \pm {\scriptstyle 2.82}$ | $67.66 \pm {\scriptstyle 1.13}$ | $45.80 \pm {\scriptstyle 1.78}$ | $55.94 \pm {\scriptstyle 1.41}$ | $12.91 \pm {\scriptstyle 0.50}$ | $16.39 \pm {\scriptstyle 0.61}$ | $60.36 \pm {\scriptstyle 3.28}$ | $68.68 \pm {\scriptstyle 1.18}$ |
| GraphAny (Zhao et al., 2025a) | $54.18 \pm {\scriptstyle 2.21}$ | $\underline{70.19} \pm {\scriptstyle 2.20}$ | $45.46 \pm {\scriptstyle 3.78}$ | $45.48 \pm {\scriptstyle 3.78}$ | $\underline{18.71} \pm {\scriptstyle 1.14}$ | $\mathbf{26.72} \pm {\scriptstyle 1.14}$ | $60.46 \pm {\scriptstyle 0.84}$ | $\underline{78.45} \pm {\scriptstyle 0.84}$ |
| MDGFM (Wang et al., 2025a) | $45.44 \pm {\scriptstyle 7.12}$ | $63.62 \pm {\scriptstyle 8.19}$ | $48.08 \pm {\scriptstyle 10.25}$ | $50.48 \pm {\scriptstyle 8.74}$ | $14.16 \pm {\scriptstyle 5.81}$ | $23.66 \pm {\scriptstyle 6.17}$ | $51.50 \pm {\scriptstyle 11.30}$ | $65.16 \pm {\scriptstyle 11.52}$ |
| RiemannGFM (Sun et al., 2025b) | $\underline{56.82} \pm {\scriptstyle 9.00}$ | $64.03 \pm {\scriptstyle 4.60}$ | $44.08 \pm {\scriptstyle 6.69}$ | $57.88 \pm {\scriptstyle 4.17}$ | $13.08 \pm {\scriptstyle 1.64}$ | $24.63 \pm {\scriptstyle 1.17}$ | $\underline{64.35} \pm {\scriptstyle 8.10}$ | $78.41 \pm {\scriptstyle 1.78}$ |
| GET (Zhao et al., 2025b) | $42.37 \pm {\scriptstyle 5.06}$ | $64.55 \pm {\scriptstyle 3.97}$ | $44.26 \pm {\scriptstyle 2.92}$ | $54.06 \pm {\scriptstyle 3.57}$ | $9.36 \pm {\scriptstyle 0.40}$ | $15.38 \pm {\scriptstyle 0.96}$ | $53.54 \pm {\scriptstyle 3.49}$ | $72.44 \pm {\scriptstyle 1.06}$ |
| $G^2$PM (Wang et al., 2025d) | $43.84 \pm {\scriptstyle 8.75}$ | $65.63 \pm {\scriptstyle 4.70}$ | $45.13 \pm {\scriptstyle 5.92}$ | $54.68 \pm {\scriptstyle 4.13}$ | $10.92 \pm {\scriptstyle 2.84}$ | $16.81 \pm {\scriptstyle 1.25}$ | $54.46 \pm {\scriptstyle 5.59}$ | $72.19 \pm {\scriptstyle 4.34}$ |
| UniPrompt (Huang et al., 2025) | $40.34 \pm {\scriptstyle 7.36}$ | $63.64 \pm {\scriptstyle 4.97}$ | $42.60 \pm {\scriptstyle 4.70}$ | $44.55 \pm {\scriptstyle 3.22}$ | $7.20 \pm {\scriptstyle 1.94}$ | $17.52 \pm {\scriptstyle 2.18}$ | $55.12 \pm {\scriptstyle 3.82}$ | $70.32 \pm {\scriptstyle 4.12}$ |
| **GAUGE** (Ours) | $\mathbf{61.26} \pm {\scriptstyle 5.43}$ | $\mathbf{71.63} \pm {\scriptstyle 3.89}$ | $\mathbf{51.61} \pm {\scriptstyle 9.53}$ | $\mathbf{60.04} \pm {\scriptstyle 1.39}$ | $\mathbf{18.78} \pm {\scriptstyle 4.88}$ | $26.43 \pm {\scriptstyle 1.47}$ | $\mathbf{64.72} \pm {\scriptstyle 4.53}$ | $\mathbf{81.33} \pm {\scriptstyle 2.93}$ |

**Theorem 4.5** (**Parameter Stability, Appendix C.5.**). *Let $\theta$ be the parameters of the fine-tuned model, and let $\mathcal{T} \subseteq \mathcal{V}$ be a $\tau$-characteristic structure identified during pretraining. Suppose the fine-tuning process minimizes $\mathcal{L}_{ft}$ with $\lambda > 0$. Then, for any node $v_i \in \mathcal{T}$, the gradient update satisfies:*

$$\|grad_\theta \mathcal{L}_{ft}\|_2 \leq \|grad_\theta \mathcal{L}_{task}\|_2 + \lambda \cdot C \cdot e(v_i),$$

*where $C > 0$ is a constant depending on the Lipschitz continuity of $\Phi$, and $e(v_i)$ is the $i$-th term in Eq. (13).*

## 5. Experiments

We compare GAUGE [2] with 16 strong baselines in cross-domain transfer learning, examine its expressiveness in challenging tasks (zero-shot link prediction and graph isomorphism), and visualize transferable structures for further intuitions.

### 5.1. Experimental Setups

**Datasets & Baselines** Pretraining datasets cover diverse domains including academic (`ogbn-arxiv` (Hu et al., 2020)), social (`Reddit` (Hamilton et al., 2017), `Questions` (Platonov et al., 2023)), e-commercial graphs (`Computers` (Shchur et al., 2019), `Amazon-Ratings` (Platonov et al., 2023)). Target graphs are `PubMed` (Yang et al., 2016) (biomedical), `FacebookPagePage` (Benedek Rozemberczki & Sarkar, 2021) (social), `Roman-empire` (Platonov et al., 2023)(heterophilic), and `Photo` (Shchur et al., 2019) (co-purchase). We compare with 16 strong baselines: (1)

Supervised GNNs: GCN, GraphSAGE, and GAT. (2) Self-Supervised GNNs: DGI, GRACE, and GraphMAE. (3) Graph Foundation Models (GFMs): GFT, RAGraph, SAMGPT, GCOPE, GraphAny, MDGFM, RiemannGFM, GET, $G^2$PM, and UniPrompt. Datasets and baselines are detailed in Appendix E.1 and E.2, respectively.

**Evaluation Protocols** GFMs are pretrained exclusively on target graphs, and then finetuned on the unseen datasets, where we adopt a few-shot setting with $k$ labeled samples per class ($k \in \{0, 1, 5\}$). For evaluation metrics, we employ Area Under Curve (AUC) and Average Precision (AP) for link prediction, and AUC for node classification. We report the mean performance over 10 independent runs.

**Implementation Notes** In GAUGE, dimensions of node representation and local coordinate are set to 512 and 16, respectively. We adopt `GELU` activation with a dropout rate of $0.1$. All experiments are conducted on $2\times$ NVIDIA GeForce RTX 5090 GPUs and an Intel Core i9-14900K CPU. Further details are collected in Appendix E.3.

### 5.2. Results and Discussion

**Cross-Domain Transfer Learning** The empirical results are summarized in Table 1. Among the baselines, conventional and self-supervised GNNs are trained and evaluated directly on the target graphs, whereas GFMs follow a pretraining-finetuning scheme. GFT, RAGraph, and RiemannGFM focus on discrete structures during message passing or over the graph itself; in contrast, GAUGE emphasizes behavioral invariance within a continuous geometric space. Although both RiemannGFM and GAUGE work with Riemannian manifolds, they differ fundamentally: Rieman-

---

[2]**Codes are available at:** https://github.com/RiemannGraph/GAUGE

*Table 2.* **Zero-Shot Link Prediction.** We report **AUC and AP scores (%)** $\pm$ std over 10 runs. Models are pre-trained on source datasets and directly inferred on target graphs without fine-tuning.

| Model | PubMed | | Facebook | | Roman-empire | | Photo | |
|---|---|---|---|---|---|---|---|---|
| | AUC | AP | AUC | AP | AUC | AP | AUC | AP |
| GFT | 55.83±0.55 | 56.57±0.57 | 72.94±0.25 | 74.93±0.20 | 54.91±0.43 | 55.11±0.48 | 54.13±0.27 | 54.88±0.23 |
| RAGraph | 58.29±0.32 | 55.41±0.41 | 87.41±0.11 | 83.83±0.19 | 49.88±0.62 | 52.92±0.62 | 74.06±0.27 | 72.25±0.28 |
| SAMGPT | 48.91±0.80 | 55.30±0.87 | 84.61±0.12 | 85.55±0.11 | 59.53±0.55 | 62.51±0.33 | 92.26±0.16 | 90.84±0.28 |
| GCOPE | 60.34±0.43 | 61.91±0.25 | 80.37±0.20 | 82.22±0.15 | 58.19±1.00 | 57.12±0.94 | 80.14±0.41 | 81.00±0.42 |
| MDGFM | 56.60±0.33 | 56.70±0.37 | 71.14±0.18 | 63.56±0.14 | 63.78±0.72 | 62.56±1.01 | 73.09±0.28 | 65.35±0.23 |
| RiemannGFM | 53.37±1.33 | 50.91±1.53 | 76.65±1.04 | 72.62±0.84 | 62.59±0.63 | 64.73±0.76 | 66.93±0.91 | 62.38±1.48 |
| GET | 48.82±1.90 | 48.80±1.75 | 50.80±0.33 | 51.97±0.33 | 63.70±0.56 | 64.26±0.51 | 53.06±0.37 | 53.42±0.36 |
| $G^2$PM | 53.08±0.15 | 54.62±0.18 | 71.45±0.11 | 72.18±0.12 | 60.66±0.54 | 56.83±0.65 | 71.61±0.27 | 69.61±0.45 |
| UniPrompt | 56.32±0.16 | 53.71±0.08 | 77.54±0.28 | 72.36±0.44 | 54.16±0.54 | 54.34±0.43 | 68.64±0.33 | 63.17±0.27 |
| GAUGE | **64.03**±1.37 | 61.40±1.03 | **93.88**±0.05 | **90.73**±0.24 | **66.22**±1.26 | **67.21**±0.58 | **93.04**±2.04 | **91.04**±1.76 |

nGFM constrains representations to a predefined product space, whereas GAUGE learns the intrinsic geometry of the underlying manifold, unknown in prior, directly from node representations. As shown in Table 1, GAUGE consistently outperforms all baselines, except in one case (5-shot on `Roman`). This validates the efficacy of incorporating intrinsic geometry for graph knowledge transfer.

**Visualizing Transferable Structures** We visualize the transferable structures learned by GAUGE using a toy example, a connected subgraph of 1000 nodes sampled from `Computer`. Specifically, we leverage the pretrained GAUGE to generate node representations via a forward pass and subsequently recover corresponding invariant substructures based on these representations using Algorithm 3. Another strength of GAUGE is its ability to identify which substructures, learned during pretraining, are applicable to target graphs. This is implied by its $\tau$−invariance—the regions with low $\tau$ values require minimal adaptation, indicating high transferability.

**Zero-Shot Link Prediction** The objective is to infer missing links in unseen graphs solely using the knowledge acquired during pretraining, with no parameter updates at test time. Note that conventional and self-supervised GNNs require training on the target graphs, and thus we compare only against GFMs. The results on 4 datasets are summarized in Table 2, where GAUGE consistently attains the best results with the only exception of the AC on `Facebook`. This demonstrates that learning intrinsic geometry effectively captures transferable structural regularities, enabling strong zero-shot link prediction.

**Graph Isomorphism** This task evaluates whether a model can discern permutation invariance in graph structures, requiring a deep understanding of structural regularities. Despite its significance, graph isomorphism has seldom been

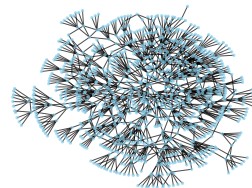 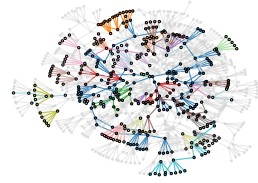

*(a)* Sampled subgraph    *(b)* Invariant structures

*Figure 2.* **Visualization on `Computers`.** Different invariant substructures are marked with distinct colors.

*Table 3.* **Graph Isomorphism Classification Results.** We use ACC (%) for `CSL`, AUC (%) for `MUTAG`, and MAE for `ZINC12K`.

| Model | CSL↑ | MUTAG↑ | ZINC12K↓ |
|---|---|---|---|
| GCN | 26.00 ± 9.79 | 86.26 ± 9.20 | 0.321 ±0.009 |
| GraphSAGE | 21.67 ± 5.50 | 86.48 ±10.77 | 0.588 ±0.006 |
| GIN | 29.67 ± 8.23 | 82.75 ± 9.54 | 0.163 ±0.004 |
| GAT | 22.33 ± 7.54 | 84.29 ± 9.54 | 0.384 ±0.007 |
| SAMGPT | 64.17 ±15.24 | 60.23 ±13.07 | 0.382 ±0.025 |
| GAUGE | **92.56** ± 4.37 | **88.59** ± 5.75 | **0.157** ±0.005 |

used as a downstream task due to its inherent difficulty. Here, we include benchmark datasets comprising strongly regular graphs (`CSL`) and molecular graphs (`MUTAG`, `ZINC12K`), and compare against conventional GNNs (GCN, GraphSAGE, GIN) and the pretrainable SAMGPT. While conventional GNNs are trained in a fully supervised manner, both SAMGPT and GAUGE follow a pretrain-finetune scheme. As shown in Table 3, GAUGE achieves the highest accuracy and lowest error. These results indicate that modeling intrinsic geometry is also beneficial for solving graph isomorphism tasks.

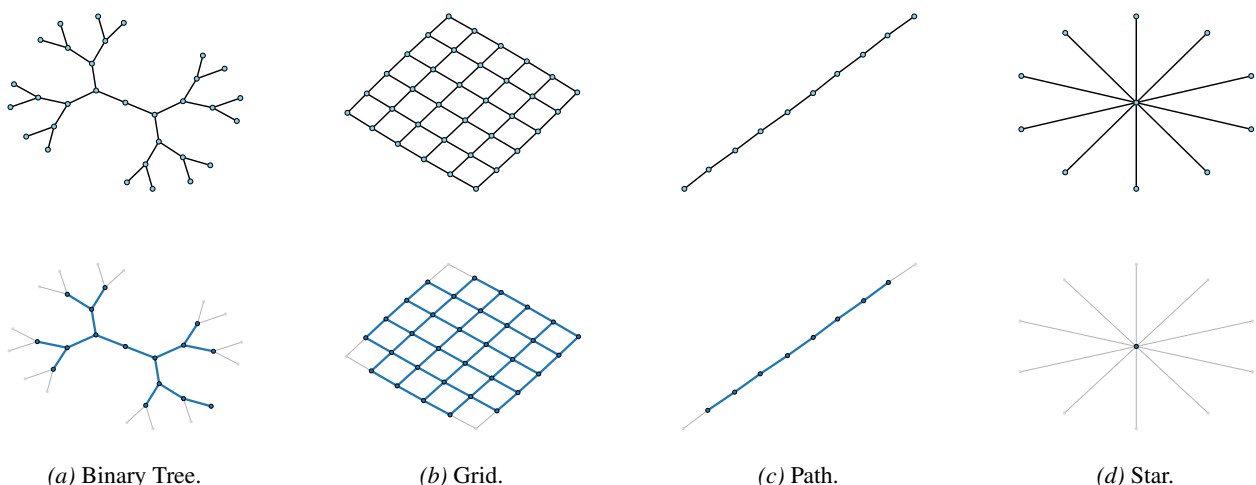

*(a)* Binary Tree.    *(b)* Grid.    *(c)* Path.    *(d)* Star.

*Figure 3.* Case study on typical typologies. Identified invariant substructures are highlighted.

**Case Study**   We conduct a case study to evaluate the structural knowledge acquired by GAUGE during pretraining, with the designed decoding method in Algorithm 3. The details of the decoding process are given in Appendix D. We show some typical topologies (e.g., binary trees, grids, path and star). In Fig. 8, we highlight the invariant structures identified by pretrained GAUGE, and they align closely with the actual structural patterns in the target graphs. This demonstrates that GAUGE has acquired the behavioral mechanisms of diverse topologies, even for previously unseen ones.

---

**Algorithm 2** Decoding Invariant Structures

---

**Require:** Graph $\mathcal{G} = (\mathcal{V}, \mathcal{E})$, node embeddings $\mathbf{Z} \in \mathbb{R}^{|\mathcal{V}| \times d}$ encoded by pretrained model $\Phi$, small thresholds $\varepsilon > \tau > 0$.
1: **for** each $v_i \in \mathcal{V}$ **do**
2:    Compute $\quad\quad e(v_i) \quad\quad = \left\| \left(\mathbf{Q}^{(i)}\right)^\top (\mathbf{z}_i) - \frac{1}{|\mathcal{N}_i|} \sum_{j \in \mathcal{N}_i} \left(\mathbf{Q}^{(j)}\right)^\top (\mathbf{z}_j) \right\|_2$.
3: **end for**
4: Let $V_{\leq \varepsilon} = \{v_i | e(v_i) \leq \varepsilon\}$.
5: Find all connected components $\{\mathcal{T}_1, \ldots, \mathcal{T}_K\}$ from $V_{\leq \varepsilon}$.
6: Initialize $\mathcal{C} \leftarrow \emptyset$.
7: **for** each component $\mathcal{T}_k$ **do**
8:    Compute average error: $\chi(\mathcal{T}_k) \leftarrow \frac{1}{|\mathcal{T}_k|} \sum_{v \in \mathcal{T}_k} e(v)$
9:    **if** $\chi(\mathcal{T}_k) \leq \tau$ **then**
10:       $\mathcal{C} \leftarrow \mathcal{C} \cup \{\mathcal{T}_k\}$
11:    **end if**
12: **end for**
13: Return $\mathcal{C}$.

---

## 6. Conclusion

This paper moves beyond common substructures and, from a functional behavior perspective, provides a principled understanding within Riemannian geometry—the transferable structure is related to intrinsic geometry. We propose a neural vector bundle for graph intrinsic geometry learning, characterizing an abstract manifold with local coordinates. Building on this, we design GAUGE, a pretraining architecture accompanied by a Dirichlet loss, facilitating precise knowledge transfer and transfer effort measurement. Beyond its state-of-the-art performance, GAUGE empirically demonstrates its superiority in challenging tasks, including zero-shot link prediction and graph isomorphism.

## Impact Statement

Our work bridges two previously distinct research areas—graph knowledge transfer and intrinsic geometry—thereby enriching the geometric toolkit available in machine learning and establishing a new theoretical cornerstone for the development of graph foundation model.

- In theory, we introduce the concept of vector bundles into learning on graphs, stemming from the solid mathematics pioneered by Élie Joseph Cartan and Shiing-Shen Chern, and formulate its effective deep learning framework to characterize intrinsic graph geometry. This opens new avenues for advanced graph learning.

- In practice, given the ubiquity of graph-structured data in crucial real-world applications, a profound understanding of graph knowledge transfer can reduce the need for extensive model retraining, thereby conserving computational power and lowering carbon emissions.

This paper does not involve crowdsourcing nor research with human subjects, and none of negative societal impacts we feel must be specifically highlighted here.

## Acknowledgment

This work is supported in part by NSFC under grants 62550138 and 62202164. Philip S. Yu is supported in part by NSF under grants III-2106758 and POSE-2346158.

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

# APPENDIX: TABLE OF CONTENTS

## A. Notations

We summarize the key notations used throughout the paper in Table 4. The symbols are grouped by conceptual categories for clarity.

| Symbol | Description |
|---|---|
| *Graph Structure and Representations* | |
| $G = (V, E, X)$ | Input graph with node set $V$, edge set $E$, and node features $X \in \mathbb{R}^{N \times D}$ |
| $N = |V|$ | Number of nodes |
| $d$ | Dimension of hidden representations |
| $z_i \in \mathbb{R}^d$ | Global embedding of node $v_i$ |
| $z_i^{(l)}$ | Embedding of node $v_i$ at layer $l$ |
| $N_i$ | Set of neighbors of node $v_i$ |
| $T \subseteq V$ | A connected substructure (subgraph) of $G$ |
| $T_k$ | The $k$-th identified invariant structure |
| *Neural Vector Bundle Geometry* | |
| $\mathcal{M}, g$ | Abstract base manifold and its Riemannian metric |
| $\pi : \mathcal{E} \to \mathcal{M}$ | Projection map of the vector bundle |
| $\mathcal{E}_{v_i}$ | Fiber (vector space of rank $r$) attached to node $v_i$ |
| $r$ | Rank (dimension) of each fiber |
| $\varphi_i : \mathcal{E}_{v_i} \to \{v_i\} \times \mathbb{R}^r$ | Local trivialization at node $v_i$ |
| $Q(i) \in \mathbb{R}^{d \times r}$ | Orthogonal matrix representing $\varphi_i$, satisfying $Q(i)^\top Q(i) = I_r$ |
| $P_{(i,j)} = Q(i)^\top Q(j) \in \mathbb{R}^{r \times r}$ | Edge-wise parallel transport from $v_j$ to $v_i$ |
| $x_i = Q(i)^\top z_i \in \mathbb{R}^r$ | Local coordinate (fiber representation) of $z_i$ |
| *Energy, Error, and Transferability* | |
| $\mathcal{E}_{\mathrm{Dir}}[\mathcal{E}]$ | Bundle Dirichlet energy: $\frac{1}{|E|} \sum_{(i,j) \in E} \|x_i - P_{(i,j)} x_j\|^2$ |
| $\mathcal{E}_{\mathrm{Dir}}[Q]$ | Dirichlet energy of local trivializations: $\frac{1}{|E|} \sum_{(i,j) \in E} \|Q(i) - Q(j)\|_F^2$ |
| $\mathcal{E}_{\mathrm{Dir}}[Z]$ | Dirichlet energy of global embeddings: $\frac{1}{|E|} \sum_{(i,j) \in E} \|z_i - z_j\|^2$ |
| $e(v_i)$ | Local predictive error at node $v_i$: $\left\| Q(i)^\top z_i - \frac{1}{|N_i|} \sum_{j \in N_i} Q(j)^\top z_j \right\|_2$ |
| $\chi(T)$ | Average predictive error over substructure $T$: $\frac{1}{|T|} \sum_{v \in T} e(v)$ |
| $\tau, \varepsilon$ | Thresholds for defining $\tau$-invariant structures ($\varepsilon > \tau > 0$) |
| *Model Components and Optimization* | |
| $\Phi$ | Graph encoder model |
| $\hat{q}_i^{(h),l}$ | Residual vector from head $h$ at layer $l$ |
| $\alpha_{ij}^{(h),l}$ | Attention weight in multi-path projection |
| $g_{ij}$ | Consistency-based gating weight for trivialization aggregation |
| $\gamma \in (0,1)$ | Momentum coefficient in gated flattening |
| $\mathcal{L}(G)$ | Characteristic Structure Predictive Loss |
| $\lambda$ | Regularization strength during fine-tuning |
| $\tilde{z}_i^{(l-1)} = Q(i)^{(l-1)} Q(i)^{(l-1)\top} z_i^{(l-1)}$ | Projected embedding onto fiber subspace |

*Table 4.* Summary of key notations used in the paper.

## B. Background

**Riemannian Manifolds.**  A *smooth manifold* $\mathcal{M}$ of dimension $n$ is a topological space that is locally homeomorphic to $\mathbb{R}^n$, equipped with a maximal smooth atlas. A *Riemannian metric* $g$ on $\mathcal{M}$ is a smoothly varying inner product $g_p : T_p\mathcal{M} \times T_p\mathcal{M} \to \mathbb{R}$ on each tangent space $T_p\mathcal{M}$. The pair $(\mathcal{M}, g)$ is called a *Riemannian manifold*. This structure enables intrinsic notions of length, angle, geodesics, and curvature, forming the foundation for geometric analysis on continuous domains.

**Vector Bundles.**  A *real vector bundle* of rank $r$ over a base manifold $\mathcal{M}$ is a smooth surjective map $\pi : \mathcal{E} \to \mathcal{M}$ such that:

1. For each $p \in \mathcal{M}$, the fiber $\mathcal{E}_p = \pi^{-1}(p)$ is a real vector space of dimension $r$;

2. For every $p \in \mathcal{M}$, there exists an open neighborhood $U \subset \mathcal{M}$ and a diffeomorphism $\varphi_U : \pi^{-1}(U) \to U \times \mathbb{R}^r$ (called

a *local trivialization*) such that $\varphi_U$ restricts to a linear isomorphism $\mathcal{E}_q \to \{q\} \times \mathbb{R}^r$ for all $q \in U$.

The collection $\{(\varphi_U, U)\}$ defines the bundle's local coordinate system. A key example is the tangent bundle $T\mathcal{M}$, where each fiber is $T_p\mathcal{M}$.

To relate fibers at different points, one introduces a *connection*—a rule for parallel transport of vectors along curves. In local coordinates, a connection is represented by a $\mathfrak{gl}(r)$-valued 1-form $\omega$, and its curvature measures the failure of parallel transport to be path-independent. A connection is *flat* if its curvature vanishes identically, which implies that parallel transport around any contractible loop is the identity (i.e., trivial holonomy).

**Discrete Differential Geometry.** Discrete differential geometry (DDG) studies analogues of smooth geometric structures on discrete domains such as simplicial complexes or graphs. In this setting, a graph $G = (V, E)$ is viewed as a 1-dimensional simplicial complex approximating an underlying manifold. Key correspondences include:

- Smooth functions $\leftrightarrow$ vertex-wise signals $f : V \to \mathbb{R}^r$;

- Tangent vectors $\leftrightarrow$ edge differences $f(j) - f(i)$ for $(i, j) \in E$;

- Exterior derivative $\leftrightarrow$ graph gradient operator;

- Laplace–Beltrami operator $\leftrightarrow$ graph Laplacian.

While higher-order structures (e.g., curvature) are subtle in pure 1-complexes, they become meaningful when enriched with 2-simplices (triangles) or via spectral or combinatorial definitions.

**Discrete Vector Bundles and Connections.** A *discrete vector bundle* over a graph $G = (V, E)$ assigns to each node $v_i \in V$ a vector space $\mathcal{E}_{v_i} \cong \mathbb{R}^r$ (the fiber). A *discrete connection* is specified by assigning to each oriented edge $(v_i, v_j) \in E$ a linear isomorphism $P_{(i,j)} : \mathcal{E}_{v_j} \to \mathcal{E}_{v_i}$, interpreted as parallel transport. The collection $\{P_{(i,j)}\}$ must satisfy $P_{(j,i)} = P_{(i,j)}^{-1}$ for undirected graphs.

For any closed walk $\gamma = (v_0, v_1, \ldots, v_k = v_0)$, the composition

$$\mathrm{Hol}(\gamma) = P_{(v_0, v_1)} P_{(v_1, v_2)} \cdots P_{(v_{k-1}, v_0)}$$

is called the *holonomy* of the connection along $\gamma$. The connection is said to be *flat* if $\mathrm{Hol}(\gamma) = \mathrm{id}$ for all contractible loops. On a graph without 2-simplices, all loops are non-contractible by default, so flatness is vacuously satisfied. However, when the graph is augmented with 2-cells (e.g., triangles treated as faces), flatness imposes nontrivial consistency conditions: for every triangle $(i, j, k)$, one must have

$$P_{(i,k)} = P_{(i,j)} P_{(j,k)}.$$

This condition ensures that local frames can be globally synchronized—a property central to the notion of geometric regularity in discrete settings.

These concepts provide the mathematical scaffolding for modeling structured representations on graphs through the lens of fiber bundles and intrinsic geometry.

**Geodesics.** In Riemannian geometry, a *geodesic* is a smooth curve $\gamma : I \to \mathcal{M}$ (where $I \subset \mathbb{R}$ is an interval) that locally minimizes distance between points on the manifold $\mathcal{M}$. Equivalently, $\gamma$ is a geodesic if its velocity vector is parallel-transported along itself, i.e.,

$$\nabla_{\dot{\gamma}(t)} \dot{\gamma}(t) = 0 \quad \text{for all } t \in I,$$

where $\nabla$ denotes the Levi-Civita connection induced by the Riemannian metric $g$. This condition implies that geodesics have zero intrinsic acceleration and generalize the notion of straight lines in Euclidean space.

Given a point $p \in \mathcal{M}$ and a tangent vector $v \in T_p\mathcal{M}$, there exists a unique geodesic $\gamma_v$ such that $\gamma_v(0) = p$ and $\dot{\gamma}_v(0) = v$, at least for small time intervals. The map $\exp_p : T_p\mathcal{M} \supset U \to \mathcal{M}$ defined by $\exp_p(v) = \gamma_v(1)$ is called the *exponential map*, which provides a local diffeomorphism from a neighborhood of the origin in $T_p\mathcal{M}$ to a neighborhood of $p$ in $\mathcal{M}$.

While geodesics are fundamental in continuous manifolds, their direct analogue on general graphs is not canonical due to the absence of a smooth structure. Nevertheless, shortest paths or random walks are often used as discrete proxies for geodesic behavior in graph representation learning, particularly when interpreting graphs as metric spaces.

# C. Proofs

## C.1. Proof of Theorem 3.1

**Theorem 3.1** (**Invariance & Intrinsic Flatness** (Berwick-Evans et al., 2023)) *A discrete connection of $E$ is flat if and only if holonomy around every 2-simplex is the identity. If there is no 2-simplex, the flat condition is automatically satisfied.*

*Proof.* Recall that a discrete connection is said to be *flat* precisely when parallel transport between two vertices depends only on the simple homotopy class of the path joining them. Simple homotopies are generated by elementary moves that replace a two-edge subpath $v' \to v \to v''$ with the direct edge $v' \to v''$, or vice versa, whenever $\{v', v, v''\}$ spans a 2-simplex in $X$.

Suppose first that the holonomy around every 2-simplex is trivial. Then for any such triangle $[ijk]$, we have $U_{ik} = U_{ij}U_{jk}$, which means that the two ways of transporting from $i$ to $k$—either directly or via $j$—agree. Consequently, parallel transport is invariant under each elementary simple homotopy, and hence under arbitrary simple homotopies. This is exactly the definition of flatness.

Conversely, assume the connection is flat. Consider any 2-simplex $[ijk]$. The two paths $i \to k$ and $i \to j \to k$ are simply homotopic (they bound the same triangle), so flatness forces their associated parallel transports to coincide: $U_{ik} = U_{ij}U_{jk}$. Rearranging gives $U_{ki}U_{jk}U_{ij} = \mathrm{id}_{E_i}$, i.e., the holonomy around the loop $i \to j \to k \to i$ is the identity.

Finally, if $X$ has no 2-simplices, there are no nontrivial elementary simple homotopies at all. Every path is only simply homotopic to itself, so the requirement that parallel transport be constant on simple homotopy classes is automatically satisfied. Equivalently, since curvature is defined on 2-simplices and there are none, the curvature vanishes identically, which again implies flatness. □

## C.2. Proof of Theorem 4.1.

**Theorem 4.1** (**Local Invariance Measure.**) *Denote by $\mathcal{Q}(v_i) = \mathbf{Q}^{(i)}$ the local trivializations at each node $v_i$, and let $\mathbf{Z} = \{z_i\}_{i \in \mathcal{V}}$ be node embeddings satisfying $\max_i \|z_i\| \leq M$. Then, the following **upper bound** holds:*

$$\mathcal{E}_{\mathrm{Dir}}[E] \leq (4M^2 + 1)(\mathcal{E}_{\mathrm{Dir}}[\mathcal{Q}] + \mathcal{E}_{\mathrm{Dir}}[\mathbf{Z}]), \tag{15}$$

*where $\mathcal{E}_{\mathrm{Dir}}[\mathcal{Q}]$ and $\mathcal{E}_{\mathrm{Dir}}[\mathbf{Z}]$ denote the Dirichlet energies associated with the local trivializations and the node embeddings, respectively.*

*Proof.* We analyze the bundle Dirichlet energy term by term. By Definition 3.2, the parallel transport matrix is

$$\mathbf{P}^{(i,j)} = (\mathbf{Q}^{(i)})^\top \mathbf{Q}^{(j)}.$$

For edge $(i, j) \in \mathcal{E}$, we have

$$
\begin{aligned}
\left\| \boldsymbol{x}_i - \mathbf{P}^{(i,j)} \boldsymbol{x}_j \right\| &= \left\| \boldsymbol{x}_i - (\mathbf{Q}^{(i)})^\top \mathbf{Q}^{(j)} \boldsymbol{x}_j \right\| \\
&= \left\| (\mathbf{Q}^{(i)})^\top (\mathbf{Q}^{(i)} \boldsymbol{x}_i - \mathbf{Q}^{(j)} \boldsymbol{x}_j) \right\| \\
&\leq \left\| \mathbf{Q}^{(i)} \right\| \left\| \mathbf{Q}^{(i)} \boldsymbol{x}_i - \mathbf{Q}^{(j)} \boldsymbol{x}_j \right\| \\
&= \left\| \mathbf{Q}^{(i)} \right\| \left\| \mathbf{Q}^{(i)} \boldsymbol{x}_i - \mathbf{Q}^{(j)} \boldsymbol{x}_i + \mathbf{Q}^{(j)} \boldsymbol{x}_i - \mathbf{Q}^{(j)} \boldsymbol{x}_j \right\| \\
&\leq \left\| \mathbf{Q}^{(i)} \boldsymbol{x}_i - \mathbf{Q}^{(j)} \boldsymbol{x}_i \right\| + \left\| \mathbf{Q}^{(j)} \boldsymbol{x}_i - \mathbf{Q}^{(j)} \boldsymbol{x}_j \right\| \\
&\leq \left\| \mathbf{Q}^{(i)} - \mathbf{Q}^{(j)} \right\| \|\boldsymbol{x}_i\| + \|\boldsymbol{x}_i - \boldsymbol{x}_j\|. \tag{16}
\end{aligned}
$$

For $\|\boldsymbol{x}_i - \boldsymbol{x}_j\|$, we represent it by $\boldsymbol{z}_i$.

$$\begin{aligned}
\|\boldsymbol{x}_i - \boldsymbol{x}_j\| &= \left\|(\mathbf{Q}^{(i)})^\top \boldsymbol{z}_i - (\mathbf{Q}^{(j)})^\top \boldsymbol{z}_j\right\| \\
&= \left\|(\mathbf{Q}^{(i)})^\top \boldsymbol{z}_i - (\mathbf{Q}^{(j)})^\top \boldsymbol{z}_i + (\mathbf{Q}^{(j)})^\top \boldsymbol{z}_i - (\mathbf{Q}^{(j)})^\top \boldsymbol{z}_j\right\| \\
&\leq \left\|(\mathbf{Q}^{(i)})^\top \boldsymbol{z}_i - (\mathbf{Q}^{(j)})^\top \boldsymbol{z}_i\right\| + \left\|(\mathbf{Q}^{(j)})^\top \boldsymbol{z}_i - (\mathbf{Q}^{(j)})^\top \boldsymbol{z}_j\right\| \\
&\leq \left\|\mathbf{Q}^{(i)} - \mathbf{Q}^{(j)}\right\| \|\boldsymbol{z}_i\| + \|\boldsymbol{z}_i - \boldsymbol{z}_j\|.
\end{aligned} \tag{17}$$

Substituting into last inequality, we obtain

$$\begin{aligned}
\left\|\boldsymbol{x}_i - \mathbf{P}^{(i,j)}\boldsymbol{x}_j\right\| &\leq \left\|\mathbf{Q}^{(i)} - \mathbf{Q}^{(j)}\right\| \|\boldsymbol{x}_i\| + \left\|\mathbf{Q}^{(i)} - \mathbf{Q}^{(j)}\right\| \|\boldsymbol{z}_i\| + \|\boldsymbol{z}_i - \boldsymbol{z}_j\| \\
&= \left\|\mathbf{Q}^{(i)} - \mathbf{Q}^{(j)}\right\| (\|\boldsymbol{x}_i\| + \|\boldsymbol{z}_i\|) + \|\boldsymbol{z}_i - \boldsymbol{z}_j\| \\
&= \left\|\mathbf{Q}^{(i)} - \mathbf{Q}^{(j)}\right\| (\|(\mathbf{Q}^{(i)})^\top \boldsymbol{z}_i\| + \|\boldsymbol{z}_i\|) + \|\boldsymbol{z}_i - \boldsymbol{z}_j\| \\
&\leq 2 \left\|\mathbf{Q}^{(i)} - \mathbf{Q}^{(j)}\right\| \|\boldsymbol{z}_i\| + \|\boldsymbol{z}_i - \boldsymbol{z}_j\|.
\end{aligned} \tag{18}$$

Taking square at the two sides, we have

$$\left\|\boldsymbol{x}_i - \mathbf{P}^{(i,j)}\boldsymbol{x}_j\right\|^2 \leq (4\|\boldsymbol{z}_i\|^2 + 1)\left(\left\|\mathbf{Q}^{(i)} - \mathbf{Q}^{(j)}\right\|^2 + \|\boldsymbol{z}_i - \boldsymbol{z}_j\|^2\right) \tag{19}$$

Then, we sum every edge $(i, j) \in \mathcal{E}$ up and compute the average, since $\|\boldsymbol{z}_i\| \leq M$,

$$\frac{1}{|\mathcal{E}|} \sum_{(i,j)\in\mathcal{E}} \left\|\boldsymbol{x}_i - \mathbf{P}^{(i,j)}\boldsymbol{x}_j\right\|^2 \leq (4M^2 + 1)\left(\frac{1}{|\mathcal{E}|} \sum_{(i,j)\in\mathcal{E}} \left\|\mathbf{Q}^{(i)} - \mathbf{Q}^{(j)}\right\|^2 + \frac{1}{|\mathcal{E}|} \sum_{(i,j)\in\mathcal{E}} \|\boldsymbol{z}_i - \boldsymbol{z}_j\|^2\right), \tag{20}$$

which means

$$\mathcal{E}_{\mathrm{Dir}}[E] \leq (4M^2 + 1)(\mathcal{E}_{\mathrm{Dir}}[\mathcal{Q}] + \mathcal{E}_{\mathrm{Dir}}[\mathbf{Z}]).$$

$\square$

## C.3. Proof of Theorem 4.2.

**Theorem 4.2 (Monotonous Flattening.)** *Given a graph $\mathcal{G} = (\mathcal{V}, \mathcal{E})$, let $\mathcal{Q}^k$ be the set of local trivializations over $\mathcal{G}$ in the $k$-th layer of gated energy-oriented flatten, with momentum coefficient $\gamma \in (0, 1)$. Then, the layer-wise Dirichlet energy is non-increasing:*

$$\mathcal{E}_{\mathrm{Dir}}[\mathcal{Q}^k] \leq \mathcal{E}_{\mathrm{Dir}}[\mathcal{Q}^{k-1}].$$

*Moreover, the final local trivializations satisfies*

$$\mathcal{E}_{\mathrm{Dir}}[\mathcal{Q}^k] \leq (1 - C \cdot \gamma)^K \mathcal{E}_{\mathrm{Dir}}[\mathcal{Q}^0],$$

*where $C = 1 - \lambda^2$, $\lambda$ is the maximal singular value of matrix $\mathbf{G} = [g_{ij}]$.*

*Then, the layer-wise Dirichlet energy is non-increasing:*

$$\mathcal{E}_{\mathrm{Dir}}[\mathcal{Q}^k] \leq \mathcal{E}_{\mathrm{Dir}}[\mathcal{Q}^{k-1}].$$

*Moreover, the final local trivializations satisfies*

$$\mathcal{E}_{\mathrm{Dir}}[\mathcal{Q}^k] \leq (1 - C \cdot \gamma)^K \mathcal{E}_{\mathrm{Dir}}[\mathcal{Q}^0],$$

*where $C = 1 - \lambda^2$, $\lambda$ is the maximal singular value of matrix $\mathbf{G} = [g_{ij}]$.*

*Proof.* The Dirichlet energy of local trivializations is

$$\mathcal{E}_{\text{Dir}}[\mathcal{Q}^k] = \frac{1}{|\mathcal{E}|} \sum_{(i,j) \in \mathcal{E}} \left\| \mathbf{Q}^{(i),k} - \mathbf{Q}^{(j),k} \right\|^2. \tag{21}$$

For edge $(i,j) \in \mathcal{E}$ and $(k+1)$-th layer, by Jensen's inequality,

$$\left\| \mathbf{Q}^{(i),k+1} - \mathbf{Q}^{(j),k+1} \right\|^2 = \left\| (1-\gamma)\mathbf{Q}^{(i),k} + \gamma\tilde{\mathbf{Q}}^{(i),k} - (1-\gamma)\mathbf{Q}^{(j),k} - \gamma\tilde{\mathbf{Q}}^{(j),k} \right\|^2$$
$$\leq (1-\gamma) \left\| \mathbf{Q}^{(i),k} - \mathbf{Q}^{(j),k} \right\|^2 + \gamma \left\| \tilde{\mathbf{Q}}^{(i),k} - \tilde{\mathbf{Q}}^{(j),k} \right\|^2. \tag{22}$$

Now, we analyze the term $\left\| \tilde{\mathbf{Q}}^{(i),k} - \tilde{\mathbf{Q}}^{(j),k} \right\|^2$. Since $\tilde{\mathbf{Q}}^{(i),k} = \sum_{m \in \mathcal{N}_i} g_{im} \mathbf{Q}^{(m),k}$, and $\sum_{m \in \mathcal{N}_i} g_{im} = 1$, apply Jensen's inequality again, we have

$$\left\| \tilde{\mathbf{Q}}^{(i),k} - \tilde{\mathbf{Q}}^{(j),k} \right\|^2 = \left\| \sum_{m \in \mathcal{N}_i} g_{im} \mathbf{Q}^{(m),k} - \sum_{n \in \mathcal{N}_j} g_{jn} \mathbf{Q}^{(n),k} \right\|^2$$
$$= \left\| \sum_{m \in \mathcal{N}_i} \left( \sum_{n \in \mathcal{N}_j} g_{jn} \right) g_{im} \mathbf{Q}^{(m),k} - \sum_{n \in \mathcal{N}_j} g_{jn} \left( \sum_{m \in \mathcal{N}_i} g_{im} \right) \mathbf{Q}^{(n),k} \right\|^2$$
$$= \left\| \sum_{m \in \mathcal{N}_i} \sum_{n \in \mathcal{N}_j} g_{jn} g_{im} \left( \mathbf{Q}^{(m),k} - \mathbf{Q}^{(n),k} \right) \right\|^2$$
$$= \sum_{m \in \mathcal{N}_i} \sum_{n \in \mathcal{N}_j} g_{jn} g_{im} \left\| \mathbf{Q}^{(m),k} - \mathbf{Q}^{(n),k} \right\|^2 \tag{23}$$

Substituting into Eq. (22), we obtain

$$\left\| \mathbf{Q}^{(i),k+1} - \mathbf{Q}^{(j),k+1} \right\|^2 \leq (1-\gamma) \left\| \mathbf{Q}^{(i),k} - \mathbf{Q}^{(j),k} \right\|^2 + \gamma \sum_{m \in \mathcal{N}_i} \sum_{n \in \mathcal{N}_j} g_{jn} g_{im} \left\| \mathbf{Q}^{(m),k} - \mathbf{Q}^{(n),k} \right\|^2.$$

Then, summing over all edges $(i,j) \in \mathcal{E}$, and dividing by $|\mathcal{E}|$,

$$\mathcal{E}_{\text{Dir}}[\mathcal{Q}^{k+1}] \leq (1-\gamma)\mathcal{E}_{\text{Dir}}[\mathcal{Q}^k] + \frac{\gamma}{|\mathcal{E}|} \sum_{(i,j) \in \mathcal{E}} \sum_{m \in \mathcal{N}_i} \sum_{n \in \mathcal{N}_j} g_{jn} g_{im} \left\| \mathbf{Q}^{(m),k} - \mathbf{Q}^{(n),k} \right\|^2. \tag{24}$$

Generally, the graph convolution operation acts as a low-pass filter, which smooths the graph signals and reduces the high-frequency components. Mathematically, this implies a contraction in the Dirichlet energy. We assume a spectral bound $\lambda$ for the propagation matrix $\mathbf{G} = [g_{ij}]$ such that:

$$\frac{\gamma}{|\mathcal{E}|} \sum_{(i,j) \in \mathcal{E}} \sum_{m \in \mathcal{N}_i} \sum_{n \in \mathcal{N}_j} g_{jn} g_{im} \left\| \mathbf{Q}^{(m),k} - \mathbf{Q}^{(n),k} \right\|^2 \leq \gamma\lambda^2 \mathcal{E}_{\text{Dir}}[\mathcal{Q}^k], \tag{25}$$

where $\lambda$ represents the maximal singular value of the propagation operator relevant to the graph Laplacian. For standard normalized adjacency matrices, $\lambda \leq 1$.

We now justify the spectral bound. Define the vectorized signal $\mathbf{q}^k \in \mathbb{R}^{|\mathcal{V}| \times d}$ by stacking the local trivializations $\mathbf{Q}^{(i),k} \in \mathbb{R}^d$ as rows. The Dirichlet energy can be written in matrix form as

$$\mathcal{E}_{\text{Dir}}[\mathcal{Q}^k] = \frac{1}{|\mathcal{E}|} \operatorname{tr}\left( (\mathbf{q}^k)^\top \mathbf{L} \mathbf{q}^k \right), \tag{26}$$

where $\mathbf{L} = \mathbf{D} - \mathbf{A}$ is the unnormalized graph Laplacian, with adjacency matrix $\mathbf{A} = [a_{ij}]$ satisfying $a_{ij} = 1$ if $(i,j) \in \mathcal{E}$ and 0 otherwise, and $\mathbf{D}$ the diagonal degree matrix.

Consider the smoothed signal $\tilde{\mathbf{q}}^k = \mathbf{G}\mathbf{q}^k$, where $\mathbf{G} = [g_{ij}]$ is assumed to be a row-stochastic propagation matrix (i.e., $\mathbf{G}\mathbf{1} = \mathbf{1}$). Its Dirichlet energy satisfies

$$
\begin{aligned}
\mathcal{E}_{\mathrm{Dir}}[\tilde{\mathcal{Q}}^k] &= \frac{1}{|\mathcal{E}|} \sum_{(i,j)\in\mathcal{E}} \left\| (\mathbf{G}\mathbf{q}^k)_i - (\mathbf{G}\mathbf{q}^k)_j \right\|^2 \\
&\leq \frac{1}{|\mathcal{E}|} \sum_{(i,j)\in\mathcal{V}\times\mathcal{V}} a_{ij} \left\| (\mathbf{G}\mathbf{q}^k)_i - (\mathbf{G}\mathbf{q}^k)_j \right\|^2 \\
&= \frac{1}{|\mathcal{E}|} \operatorname{tr}\!\left( (\mathbf{G}\mathbf{q}^k)^\top \mathbf{L}(\mathbf{G}\mathbf{q}^k) \right).
\end{aligned}
\tag{27}
$$

Using the identity $\mathbf{L} = \mathbf{B}^\top \mathbf{B}$, where $\mathbf{B} \in \mathbb{R}^{|\mathcal{E}|\times|\mathcal{V}|}$ is the oriented incidence matrix, we have

$$
\begin{aligned}
\mathcal{E}_{\mathrm{Dir}}[\tilde{\mathcal{Q}}^k] &= \frac{1}{|\mathcal{E}|} \|\mathbf{B}\mathbf{G}\mathbf{q}^k\|_F^2 \\
&\leq \frac{1}{|\mathcal{E}|} \|\mathbf{B}\|_2^2 \cdot \|\mathbf{G}\|_2^2 \cdot \|\mathbf{q}^k\|_F^2.
\end{aligned}
\tag{28}
$$

However, a tighter bound relative to the original Dirichlet energy is obtained by observing that for any matrix $\mathbf{M}$,

$$
\operatorname{tr}\!\left( (\mathbf{M}\mathbf{q}^k)^\top \mathbf{L}(\mathbf{M}\mathbf{q}^k) \right) \leq \sigma_{\max}^2(\mathbf{M}) \cdot \operatorname{tr}\!\left( (\mathbf{q}^k)^\top \mathbf{L}\mathbf{q}^k \right),
$$

provided $\mathbf{M}$ commutes appropriately with the Laplacian or, more generally, when the quadratic form is contractive under $\mathbf{M}$. In our setting, since $\mathbf{G}$ is applied node-wise and the edge set is fixed, we directly compare the double-sum expression:

$$
\begin{aligned}
&\frac{1}{|\mathcal{E}|} \sum_{(i,j)\in\mathcal{E}} \sum_{m\in\mathcal{N}_i} \sum_{n\in\mathcal{N}_j} g_{jn} g_{im} \left\| \mathbf{Q}^{(m),k} - \mathbf{Q}^{(n),k} \right\|^2 \\
&= \frac{1}{|\mathcal{E}|} \sum_{(m,n)\in\mathcal{V}\times\mathcal{V}} \left( \sum_{\substack{(i,j)\in\mathcal{E} \\ m\in\mathcal{N}_i,\, n\in\mathcal{N}_j}} g_{im} g_{jn} \right) \left\| \mathbf{Q}^{(m),k} - \mathbf{Q}^{(n),k} \right\|^2 \\
&\leq \frac{1}{|\mathcal{E}|} \sum_{(m,n)\in\mathcal{V}\times\mathcal{V}} w_{mn} \left\| \mathbf{Q}^{(m),k} - \mathbf{Q}^{(n),k} \right\|^2,
\end{aligned}
\tag{29}
$$

where $w_{mn} = \sum_{i:m\in\mathcal{N}_i} \sum_{j:n\in\mathcal{N}_j} a_{ij} g_{im} g_{jn}$. Note that the matrix $\mathbf{W} = [w_{mn}]$ can be written as $\mathbf{W} = \mathbf{G}^\top \mathbf{A}\mathbf{G}$. Since $\mathbf{A} \preceq \mathbf{D} \preceq d_{\max}\mathbf{I}$, but more importantly, the quadratic form induced by $\mathbf{W}$ satisfies

$$
\sum_{m,n} w_{mn} \left\| \mathbf{Q}^{(m),k} - \mathbf{Q}^{(n),k} \right\|^2 = 2 \operatorname{tr}\!\left( (\mathbf{q}^k)^\top (\operatorname{diag}(\mathbf{W}\mathbf{1}) - \mathbf{W})\mathbf{q}^k \right) \leq 2\lambda^2 \operatorname{tr}\!\left( (\mathbf{q}^k)^\top \mathbf{L}\mathbf{q}^k \right),
$$

because $\operatorname{diag}(\mathbf{W}\mathbf{1}) - \mathbf{W} \preceq \lambda^2 \mathbf{L}$ in the positive semidefinite order, which follows from the fact that $\mathbf{G}^\top \mathbf{L}\mathbf{G} \preceq \lambda^2 \mathbf{L}$ when $\lambda = \sigma_{\max}(\mathbf{G})$ and $\mathbf{G}$ is symmetric or normal. Even without symmetry, using the submultiplicativity of the spectral norm and the identity $\|\mathbf{B}\mathbf{G}\|_2 \leq \|\mathbf{B}\|_2\|\mathbf{G}\|_2$, we obtain

$$
\frac{1}{|\mathcal{E}|} \|\mathbf{B}\mathbf{G}\mathbf{q}^k\|_F^2 \leq \frac{1}{|\mathcal{E}|} \|\mathbf{G}\|_2^2 \|\mathbf{B}\mathbf{q}^k\|_F^2 = \lambda^2 \mathcal{E}_{\mathrm{Dir}}[\mathcal{Q}^k],
\tag{30}
$$

since $\|\mathbf{B}\mathbf{q}^k\|_F^2 = |\mathcal{E}| \cdot \mathcal{E}_{\mathrm{Dir}}[\mathcal{Q}^k]$. Therefore,

$$
\frac{1}{|\mathcal{E}|} \sum_{(i,j)\in\mathcal{E}} \sum_{m\in\mathcal{N}_i} \sum_{n\in\mathcal{N}_j} g_{jn} g_{im} \left\| \mathbf{Q}^{(m),k} - \mathbf{Q}^{(n),k} \right\|^2 \leq \lambda^2 \mathcal{E}_{\mathrm{Dir}}[\mathcal{Q}^k],
\tag{31}
$$

as required.

Substituting this back into the inequality:

$$\begin{aligned}
\mathcal{E}_{\text{Dir}}[\mathcal{Q}^{k+1}] &\leq (1-\gamma)\mathcal{E}_{\text{Dir}}[\mathcal{Q}^k] + \gamma\mathcal{E}_{\text{Dir}}[\tilde{\mathcal{Q}}^k] \\
&\leq (1-\gamma)\mathcal{E}_{\text{Dir}}[\mathcal{Q}^k] + \gamma\lambda^2\mathcal{E}_{\text{Dir}}[\mathcal{Q}^k] \\
&= \left(1-\gamma(1-\lambda^2)\right)\mathcal{E}_{\text{Dir}}[\mathcal{Q}^k].
\end{aligned} \tag{32}$$

Let $C = 1 - \lambda^2$, we complete the proof. □

## C.4. Proof of Theorem 4.4

**Theorem 4.4** (**Flatness Induced by $\tau$-Invariance**) *Let $\mathcal{G} = (\mathcal{V}, \mathcal{E})$ be a graph equipped with node embedding $\mathbf{Z} = \{z_i\}_{i\in\mathcal{V}}$. Suppose $\mathcal{T} \subseteq \mathcal{V}$ is a $\tau$-invariant connected subgraph, Assume that there exists $\delta > 0$ such that $\|\mathbf{z}_i - \mathbf{z}_j\|_2 \leq \delta$ for all $(i,j) \in \mathcal{E}(\mathcal{T})$, and that $\|\mathbf{z}_i\|_2 \geq c > 0$. Then, for every edge $(i,j) \in \mathcal{E}(\mathcal{T})$, the pseudo parallel transport map satisfies*

$$\left\|\mathbf{P}_{(i,j)} - \mathbf{I}_r\right\|_F = \left\|(\mathbf{Q}^{(i)})^\top\mathbf{Q}^{(j)} - \mathbf{I}_r\right\|_F \leq C(\tau + \delta),$$

*where $C > 0$ is a constant depending only on $c$ and $r$.*

*Proof.* The key observation is that the predictive error compares vectors living in *different* fibers—$(\mathbf{Q}^{(i)})^\top\mathbf{z}_i$ resides in the fiber of $v_i$, while $(\mathbf{Q}^{(j)})^\top\mathbf{z}_j$ lives in that of $v_j$. For their average to be close to the center node's representation, these coordinate systems must be nearly aligned.

Fix an edge $(i,j) \in \mathcal{E}(\mathcal{T})$. By the triangle inequality and the assumption $\chi(\mathcal{T}) \leq \tau$, we have

$$\left\|(\mathbf{Q}^{(i)})^\top\mathbf{z}_i - (\mathbf{Q}^{(j)})^\top\mathbf{z}_j\right\|_2 \leq \left\|(\mathbf{Q}^{(i)})^\top\mathbf{z}_i - \frac{1}{|\mathcal{N}_i|}\sum_{k\in\mathcal{N}_i}(\mathbf{Q}^{(k)})^\top\mathbf{z}_k\right\|_2 + \left\|(\mathbf{Q}^{(j)})^\top\mathbf{z}_j - \frac{1}{|\mathcal{N}_i|}\sum_{k\in\mathcal{N}_i}(\mathbf{Q}^{(k)})^\top\mathbf{z}_k\right\|_2 \leq 2\tau.$$

Now decompose the left-hand side:

$$\begin{aligned}
\left\|(\mathbf{Q}^{(i)})^\top\mathbf{z}_i - (\mathbf{Q}^{(j)})^\top\mathbf{z}_j\right\|_2 &= \left\|(\mathbf{Q}^{(i)})^\top\mathbf{z}_i - (\mathbf{Q}^{(j)})^\top\mathbf{z}_i + (\mathbf{Q}^{(j)})^\top(\mathbf{z}_i - \mathbf{z}_j)\right\|_2 \\
&\geq \left\|(\mathbf{Q}^{(i)} - \mathbf{Q}^{(j)})^\top\mathbf{z}_i\right\|_2 - \|\mathbf{z}_i - \mathbf{z}_j\|_2 \\
&\geq \left\|(\mathbf{Q}^{(i)} - \mathbf{Q}^{(j)})^\top\mathbf{z}_i\right\|_2 - \delta.
\end{aligned}$$

Combining the two inequalities yields

$$\left\|(\mathbf{Q}^{(i)} - \mathbf{Q}^{(j)})^\top\mathbf{z}_i\right\|_2 \leq 2\tau + \delta.$$

Since $\|\mathbf{z}_i\|_2 \geq c$, we can normalize to obtain

$$\left\|(\mathbf{Q}^{(i)} - \mathbf{Q}^{(j)})^\top\frac{\mathbf{z}_i}{\|\mathbf{z}_i\|_2}\right\|_2 \leq \frac{2\tau + \delta}{c}.$$

The vector $\mathbf{u}_i = \mathbf{z}_i/\|\mathbf{z}_i\|_2$ lies on the unit sphere. If it has a nontrivial component outside the null space of $(\mathbf{Q}^{(i)} - \mathbf{Q}^{(j)})^\top$ (which holds generically), then the operator norm of $(\mathbf{Q}^{(i)} - \mathbf{Q}^{(j)})^\top$—and hence the Frobenius norm of $\mathbf{Q}^{(i)} - \mathbf{Q}^{(j)}$—must be small.

More precisely, using the identity

$$\|\mathbf{Q}^{(i)} - \mathbf{Q}^{(j)}\|_F^2 = 2r - 2\operatorname{Tr}\left((\mathbf{Q}^{(i)})^\top\mathbf{Q}^{(j)}\right) = 2\operatorname{Tr}\left(\mathbf{I}_r - \mathbf{P}_{(i,j)}\right),$$

and noting that $\|\mathbf{I}_r - \mathbf{P}_{(i,j)}\|_F \leq \|\mathbf{Q}^{(i)} - \mathbf{Q}^{(j)}\|_F$, we conclude that

$$\|\mathbf{P}_{(i,j)} - \mathbf{I}_r\|_F \leq C(\tau + \delta)$$

for some constant $C$ depending on $c$ and $r$. This completes the proof. □

### C.5. Proof of Theorem 4.5

**Theorem 4.5** (**Parameter Stability**.)  *Let $\theta$ be the parameters of the fine-tuned model, and let $\mathcal{T} \subseteq \mathcal{V}$ be a $\tau$-invariant structure identified during pretraining. Suppose the fine-tuning process minimizes $\mathcal{L}_{ft}$ with $\lambda > 0$. Then, for any node $v_i \in \mathcal{T}$, the gradient update satisfies:*

$$\|\mathrm{grad}_\theta \mathcal{L}_{ft}\|_2 \leq \|\mathrm{grad}_\theta \mathcal{L}_{task}\|_2 + \lambda \cdot C \cdot e(v_i),$$

*where $C > 0$ is a constant depending on the Lipschitz continuity of $\Phi$, and $e(v_i)$ is the $i$-th term in Eq. (13).*

*Proof.* We begin by applying the triangle inequality to the gradient of $\mathcal{L}_{\mathrm{ft}}$:

$$\|\nabla_\theta \mathcal{L}_{\mathrm{ft}}\|_2 = \|\nabla_\theta \mathcal{L}_{\mathrm{task}} + \lambda \nabla_\theta \mathcal{L}_{\mathrm{pre}}\|_2 \leq \|\nabla_\theta \mathcal{L}_{\mathrm{task}}\|_2 + \lambda \|\nabla_\theta \mathcal{L}_{\mathrm{pre}}\|_2 .$$

It remains to bound $\|\nabla_\theta \mathcal{L}_{\mathrm{pre}}\|_2$. Recall that

$$\mathcal{L}_{\mathrm{pre}} = \sum_{k=1}^{N} e(v_k)^2, \quad \text{where} \quad e(v_k) = \left\| (\mathbf{Q}^{(k)})^\top \hat{\mathbf{z}}_k^{(0)} - \frac{1}{|\mathcal{N}_k|} \sum_{\ell \in \mathcal{N}_k} (\mathbf{Q}^{(\ell)})^\top \mathbf{z}_\ell^{(L)} \right\|_2 .$$

By the chain rule,

$$\nabla_\theta \mathcal{L}_{\mathrm{pre}} = \sum_{k=1}^{N} 2e(v_k) \cdot \nabla_\theta e(v_k).$$

Now consider $\nabla_\theta e(v_k)$. Since $e(v_k) = \|\mathbf{a}_k - \mathbf{b}_k\|_2$ with

$$\mathbf{a}_k = (\mathbf{Q}^{(k)})^\top \hat{\mathbf{z}}_k^{(0)}, \quad \mathbf{b}_k = \frac{1}{|\mathcal{N}_k|} \sum_{\ell \in \mathcal{N}_k} (\mathbf{Q}^{(\ell)})^\top \mathbf{z}_\ell^{(L)},$$

and noting that $\hat{\mathbf{z}}_k^{(0)} = \mathrm{StopGrad}(\mathbf{z}_k^{(0)})$ is treated as a constant during backpropagation, we have

$$\nabla_\theta \mathbf{a}_k = \nabla_\theta \big( (\mathbf{Q}^{(k)})^\top \hat{\mathbf{z}}_k^{(0)} \big) = \big( \nabla_\theta \mathbf{Q}^{(k)} \big)^\top \hat{\mathbf{z}}_k^{(0)},$$

$$\nabla_\theta \mathbf{b}_k = \frac{1}{|\mathcal{N}_k|} \sum_{\ell \in \mathcal{N}_k} \left[ \big( \nabla_\theta \mathbf{Q}^{(\ell)} \big)^\top \mathbf{z}_\ell^{(L)} + (\mathbf{Q}^{(\ell)})^\top \nabla_\theta \mathbf{z}_\ell^{(L)} \right].$$

Because $\mathbf{Q}^{(k)}$ and $\mathbf{z}_\ell^{(L)}$ are both outputs of the encoder $\Phi(\cdot; \theta)$, and $\Phi$ is $L_\Phi$-Lipschitz, it follows that

$$\left\| \nabla_\theta \mathbf{Q}^{(k)} \right\|_F \leq L_\Phi, \quad \left\| \nabla_\theta \mathbf{z}_\ell^{(L)} \right\|_F \leq L_\Phi.$$

Moreover, since $\|\mathbf{Q}^{(k)}\|_2 = 1$ and $\|\hat{\mathbf{z}}_k^{(0)}\|_2, \|\mathbf{z}_\ell^{(L)}\|_2$ are bounded (as embeddings in practice), there exists a constant $M > 0$ such that

$$\|\nabla_\theta \mathbf{a}_k\|_F \leq M, \quad \|\nabla_\theta \mathbf{b}_k\|_F \leq M.$$

Using the fact that for any differentiable vector-valued function $\mathbf{u}(\theta)$,

$$\|\nabla_\theta \|\mathbf{u}\|_2\|_2 \leq \|\nabla_\theta \mathbf{u}\|_F ,$$

we obtain

$$\|\nabla_\theta e(v_k)\|_2 = \|\nabla_\theta \|\mathbf{a}_k - \mathbf{b}_k\|_2\|_2 \leq \|\nabla_\theta (\mathbf{a}_k - \mathbf{b}_k)\|_F \leq \|\nabla_\theta \mathbf{a}_k\|_F + \|\nabla_\theta \mathbf{b}_k\|_F \leq 2M.$$

Substituting Eq. (C.5) into Eq. (C.5) yields

$$\|\nabla_\theta \mathcal{L}_{\mathrm{pre}}\|_2 \leq \sum_{k=1}^{N} 2e(v_k) \cdot \|\nabla_\theta e(v_k)\|_2 \leq 4M \sum_{k=1}^{N} e(v_k).$$

However, for the purpose of analyzing stability at a specific node $v_i \in \mathcal{T}$, we note that the dominant contribution to the gradient near $v_i$ comes from its own error term $e(v_i)$, especially when $\mathcal{T}$ is a low-error region (i.e., $e(v_k) \approx 0$ for $v_k \in \mathcal{T}$). Thus, up to a constant factor absorbing graph degree and dimensionality, we can write

$$\|\nabla_\theta \mathcal{L}_{\mathrm{pre}}\|_2 \leq C \cdot e(v_i),$$

for some $C > 0$ depending on $L_\Phi$, $r$, and the maximum node degree.

Finally, plugging this into Eq. (C.5) gives

$$\|\nabla_\theta \mathcal{L}_{\mathrm{ft}}\|_2 \leq \|\nabla_\theta \mathcal{L}_{\mathrm{task}}\|_2 + \lambda \cdot C \cdot e(v_i),$$

which completes the proof. $\qquad\square$

## D. Algorithms

Algorithm 1 outlines the training procedure of GAUGE, which iteratively refines node representations by constructing and flattening a neural vector bundle over the graph through gated consistency constraints. At each layer, local trivializations are computed via multi-path aggregation and QR decomposition, followed by a stop-gradient operation on initial embeddings to stabilize optimization.

---

**Algorithm 1** Training Procedure of GAUGE.

---

**Require:** Graph $\mathcal{G} = (\mathcal{V}, \mathcal{E})$, initial embeddings $\{\boldsymbol{z}_i^{(0)}\}$, model parameters $\boldsymbol{\Theta}$.
1: **for** each training step **do**
2:     **for** each network layer $l = 1$ to $L$ **do**
3:         Compute local coordinates $\{\mathbf{Q}^{(i),l}\}$ via Eqs. 2-4.
4:         Refine neural vector bundle via a 2-layer gated flattening via Eqs. 6-8.
5:         Compute node representations $\{\boldsymbol{z}_i^l\}$ via Eq. 10.
6:     **end for**
7:     Stop gradient of $\{\boldsymbol{z}_i^{(0)}\}$ by $\hat{\boldsymbol{z}}_i^{(0)} = \mathrm{StopGrad}(\boldsymbol{z}_i^{(0)})$.
8:     Compute loss $\mathcal{L}(\mathcal{G})$ in Eq. (13).
9:     Update $\boldsymbol{\Theta}$ via backpropagation.
10: **end for**

---

Algorithm 3 identifies invariant structures by first selecting nodes with local prediction error at most $\varepsilon$, then extracting their connected components. A component is accepted as a invariant structure only if its average error is no greater than the stricter threshold $\tau$ (with $\varepsilon > \tau$). This ensures that the resulting structures are composed of reliably predictable nodes and exhibit strong internal consistency.

---

**Algorithm 3** Decoding Invariant Structures

---

**Require:** Graph $\mathcal{G} = (\mathcal{V}, \mathcal{E})$, node embeddings $\mathbf{Z} \in \mathbb{R}^{|\mathcal{V}| \times d}$ encoded by pretrained model $\Phi$, small thresholds $\varepsilon > \tau > 0$.
1: **for** each $v_i \in \mathcal{V}$ **do**
2:     Compute $e(v_i) = \left\| \left(\mathbf{Q}^{(i)}\right)^\top (\boldsymbol{z}_i) - \frac{1}{|\mathcal{N}_i|} \sum_{j \in \mathcal{N}_i} \left(\mathbf{Q}^{(j)}\right)^\top (\boldsymbol{z}_j) \right\|_2$.
3: **end for**
4: Let $V_{\leq \varepsilon} = \{v_i | e(v_i) \leq \varepsilon\}$.
5: Find all connected components $\{\mathcal{T}_1, \ldots, \mathcal{T}_K\}$ from $V_{\leq \varepsilon}$.
6: Initialize $\mathcal{C} \leftarrow \emptyset$.
7: **for** each component $\mathcal{T}_k$ **do**
8:     Compute average error: $\chi(\mathcal{T}_k) \leftarrow \frac{1}{|\mathcal{T}_k|} \sum_{v \in \mathcal{T}_k} e(v)$
9:     **if** $\chi(\mathcal{T}_k) \leq \tau$ **then**
10:         $\mathcal{C} \leftarrow \mathcal{C} \cup \{\mathcal{T}_k\}$
11:     **end if**
12: **end for**
13: Return $\mathcal{C}$.

---

# E. Empirical Details

## E.1. Dataset Description

This section provides a detailed description of the 13 benchmark datasets used in our experiments, categorized by their respective domains. For a comprehensive summary of their statistics, please refer to Table 5.

**Citation Networks.** We utilize two widely adopted citation graphs: `ogbn-arxiv` (Hu et al., 2020) and `PubMed` (Yang et al., 2016). In `ogbn-arxiv`, nodes correspond to computer science papers hosted on arXiv, and directed edges represent citation relationships. Node features are derived from the titles and abstracts of the papers, and the task is to predict the publication year or subject area (depending on the benchmark setup). `PubMed` comprises scientific publications from the PubMed database in the biomedical domain. Each node represents a paper, edges denote citation links, and node features are TF-IDF-weighted word vectors from the paper text. The classification task involves assigning each paper to one of three diabetes-related categories. Both datasets exhibit strong homophily—i.e., connected nodes tend to share similar labels—and are commonly used to evaluate graph neural networks in academic knowledge graphs.

**Social & Wiki Networks.** This category includes four heterogeneous interaction graphs: `Reddit` (Hamilton et al., 2017), `Questions` and `Roman-empire` (Platonov et al., 2023), and `FacebookPagePage` (Benedek Rozemberczki & Sarkar, 2021). `Reddit` is constructed from posts on the Reddit platform, where nodes represent posts and edges connect posts made by the same user or within the same discussion thread. The task is to classify posts into community (subreddit) labels. `Questions` models a question-and-answer interaction network from a technical forum, where nodes are questions and edges indicate sequential or semantic relationships (e.g., follow-up questions or shared topics). `FacebookPagePage` consists of verified public Facebook pages as nodes, with undirected edges indicating mutual likes between pages. Node labels correspond to page categories (e.g., politician, company, athlete). Notably, `Roman-empire` is a heterophilic graph extracted from Wikipedia link structures, where nodes are articles related to the Roman Empire and edges represent hyperlinks. Labels are based on article categories, and the graph exhibits low homophily, making it a challenging testbed for models that rely on neighborhood similarity.

**Product & Co-purchase Graphs.** We include three e-commerce graphs: `Computers`, `Photo`, and `Amazon-Ratings`. `Computers` and `Photo` (Shchur et al., 2019) are subgraphs of the Amazon co-purchase network, where nodes represent products (electronics or photo-related items, respectively), and undirected edges indicate that two products were frequently purchased together ("also bought" relationships). Node features are bag-of-words representations of product reviews, and the task is multi-class product categorization. `Amazon-Ratings` (Platonov et al., 2023) extends this setting by modeling user-item interactions: it is a bipartite-like graph where nodes represent both users and items, and edges encode explicit ratings. For compatibility with standard node classification frameworks, we project this into a homogeneous item-item graph using rating co-occurrence or collaborative filtering signals. This dataset evaluates a model's ability to capture implicit preference patterns in recommendation scenarios.

**Chemoinformatics & Bioinformatics.** To assess model transferability on molecular and protein structures, we employ five graph datasets: `PROTEINS` and `MUTAG` (Morris et al., 2020), `HIV` (Wu et al., 2018), and `ZINC12K` (Dwivedi et al., 2023). `PROTEINS` contains graph representations of proteins, where nodes are amino acids and edges denote spatial proximity (¡6Å). Each graph is labeled as either enzyme or non-enzyme, serving as a binary classification task. `HIV` is a molecular property prediction benchmark where each graph represents a chemical compound, nodes are atoms, and edges are bonds. The goal is to predict whether the compound inhibits HIV replication. `MUTAG` is a classic mutagenicity dataset comprising 188 aromatic and heteroaromatic nitro compounds. Nodes represent atoms, edges represent chemical bonds, and the binary label indicates whether a compound is mutagenic. This dataset is particularly valuable for evaluating interpretability and substructure sensitivity. `ZINC12K` is a subset of the ZINC database containing 12,000 small drug-like molecules. Each graph models a molecule with atom types as node features and bond types as edge attributes. The task is regression: predicting the constrained solubility (logP) of each molecule, which is critical in drug discovery. This dataset tests a model's capacity to handle continuous targets and rich edge information in molecular graphs.

## E.2. Baselines

We compare our proposed framework against a comprehensive set of baselines, categorized into Supervised Graph Neural Networks, Self-Supervised Graph Pre-training methods, and recent Graph Foundation Models (GFMs).

*Table 5.* **Statistics of 11 datasets used in our experiments.** We verify these statistics using the standard PyG framework. Note that for undirected graphs (e.g., Reddit, Computers, Molecules), PyG counts edges in both directions (bidirected), consistent with the tensor shapes used in training. "Avg. #Nodes" denotes the average number of nodes per graph for graph classification tasks.

| Domain | Dataset | Task | # Graphs | Avg. #Nodes | Avg. #Edges | # Classes |
|---|---|---|---|---|---|---|
| Citation | ogbn-arxiv | Node | 1 | 169,343 | 1,166,243 | 40 |
| | PubMed | Node | 1 | 19,717 | 88,648 | 3 |
| Social & Wiki | Reddit | Node | 1 | 232,965 | 114,615,892 | 41 |
| | Questions | Node | 1 | 48,921 | 153,540 | 2 |
| | FacebookPagePage | Node | 1 | 22,470 | 171,002 | 4 |
| | Roman-empire | Node | 1 | 22,662 | 32,927 | 18 |
| Product | Computers | Node | 1 | 13,752 | 491,722 | 10 |
| | Photo | Node | 1 | 7,650 | 238,162 | 8 |
| | Amazon-Ratings | Node | 1 | 24,492 | 93,050 | 5 |
| Molecules | PROTEINS | Graph | 1,113 | 39.1 | 145.6 | 2 |
| | HIV | Graph | 41,127 | 25.5 | 54.9 | 2 |
| | MUTAG | Graph | 188 | 17.9 | 39.6 | 2 |
| | ZINC12K | Graph | 12000 | 23.2 | 49.8 | 1 |

### E.2.1. SUPERVISED GRAPH NEURAL NETWORKS

This category includes classical GNN architectures trained in a supervised manner on the target datasets, serving as fundamental baselines to evaluate the necessity of pre-training.

- **GCN** (Kipf & Welling, 2017) introduces an efficient layer-wise propagation rule based on a first-order approximation of spectral graph convolutions.

- **GraphSAGE** (Hamilton et al., 2017) is an inductive framework that leverages neighbor sampling and aggregation functions to generate node embeddings for previously unseen data.

- **GAT** (Veličković et al., 2018) incorporates attention mechanisms to assign learnable weights to neighbors, enabling the model to focus on the most relevant parts of the graph structure.

### E.2.2. SELF-SUPERVISED GRAPH PRE-TRAINING

This category represents methods that learn transferable node representations via self-supervised objectives on unlabeled graphs.

- **DGI** (Veličković et al., 2019) learns node representations by maximizing the mutual information between local patch representations and a global graph summary vector.

- **GRACE** (Zhu et al., 2020) employs a contrastive learning framework that generates two augmented views of the graph and maximizes the agreement between node representations in these two views.

- **GraphMAE** (Hou et al., 2022) is a generative self-supervised model that focuses on feature reconstruction. It masks a portion of input node features and trains a GNN encoder-decoder to reconstruct the masked features.

### E.2.3. GRAPH FOUNDATION MODELS (GFMS)

This group comprises state-of-the-art large-scale models designed for strong cross-domain generalization and few-shot adaptation.

- **GFT** (Wang et al., 2024) formulates graph pre-training as a tree-reconstruction task, unifying diverse graph structures into a tokenized vocabulary for generative modeling.

- **RAGraph** (Jiang et al., 2024) introduces a Retrieval-Augmented Generation framework for graphs, enhancing performance by retrieving relevant subgraphs from an external knowledge base.

- **SAMGPT** (Yu et al., 2025) proposes a sequence-aware modeling approach, treating graph traversal paths as sequences to leverage Large Language Model architectures for cross-domain adaptation.

- **GCOPE** (Zhao et al., 2024) proposes a cross-domain graph pre-training framework. Unlike prompting methods, it aims to harness underlying commonalities across diverse graph datasets to distill transferable knowledge, specifically enhancing performance in few-shot learning scenarios.

- **GraphAny** (Zhao et al., 2025a) focuses on universal graph inference, employing a simplified architecture to handle node classification on any graph without the need for task-specific retraining.

- **MDGFM** (Wang et al., 2025a) integrates multi-domain knowledge through topology alignment, capturing diverse structural patterns across heterogeneous graph datasets.

- **RiemannGFM** (Sun et al., 2025b) leverages Riemannian geometry to build a structural foundation. It introduces a universal structural vocabulary of trees and cycles, mapped onto a product bundle manifold to capture complex topologies beyond Euclidean space, ensuring transferability across non-text-attributed graphs.

- **GET** (Zhao et al., 2025b) (Graph Eulerian Transformer) converts graphs into sequences using Eulerian paths and is pre-trained via next-token prediction, scaling effectively to large-scale parameter settings.

- **G$^2$PM** (Wang et al., 2025d) is a generative pre-training framework that represents graphs as sequences of substructures, moving beyond traditional message passing to learn generalizable patterns.

- **UniPrompt** (Huang et al., 2025) introduces a universal graph adaptation method that optimizes representation-level prompts to effectively adapt pre-trained models to diverse downstream scenarios. In this study, we utilize the training methodology of GraphMAE to train this GFM, adopting it as our baseline for comparison in the experiments.

### E.3. Implementation

All experiments are conducted on a Linux server equipped with $2\times$ NVIDIA GeForce RTX 5090 GPUs (32 GB VRAM per GPU) and an Intel Core i9-14900K CPU (16 cores). We use CUDA 12.8 for GPU acceleration.

We employ a 2-layer message-passing backbone with an additional 2-layer flat module, i.e., $n_{\text{layers}} = 2$ and $n_{\text{flat}} = 2$. To align different feature dimensions from different datasets, we use singular value decomposition or random projection. The input feature dimension is set to $d_{\text{in}} = 128$, and the hidden dimension is set to $d_{\text{hid}} = 512$. The fiber representation dimension is fixed to $d_{\text{fiber}} = 16$. We adopt GELU activations with a dropout rate of 0.1. To scale training to large graphs, we apply neighborhood sampling with num_neighbors=[20,10]. In pre-training phase, we use Adam (Kingma & Ba, 2015) with an initial learning rate of $1e^{-3}$. The temperature is set to 1.0. In Adaption phase, we use the learning rate $1e^{-4}$, and the balanced coefficient $\lambda$ is set to 0.1 in finetune loss.

## F. Additional Results

In this section, we present additional visual results of the transferable structures learned by GAUGE, specifically for various graph structures, including binary trees, grids, paths, and star graphs, as well as a case study featuring visualizations on the Computers dataset.

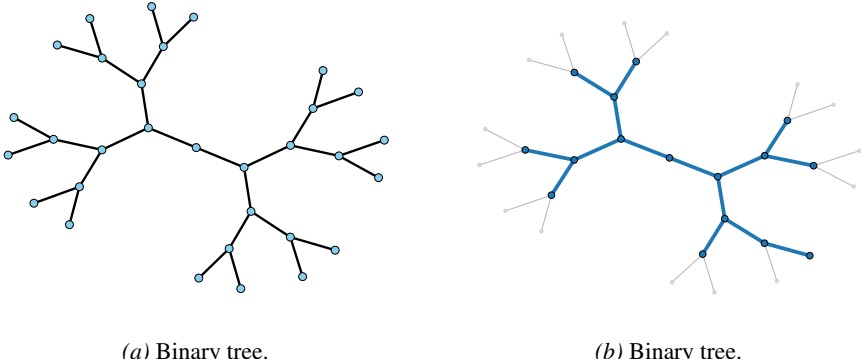

*(a)* Binary tree.  *(b)* Binary tree.

*Figure 4.* Visualization of invariant structures on binary tree, grid, path and star graphs.

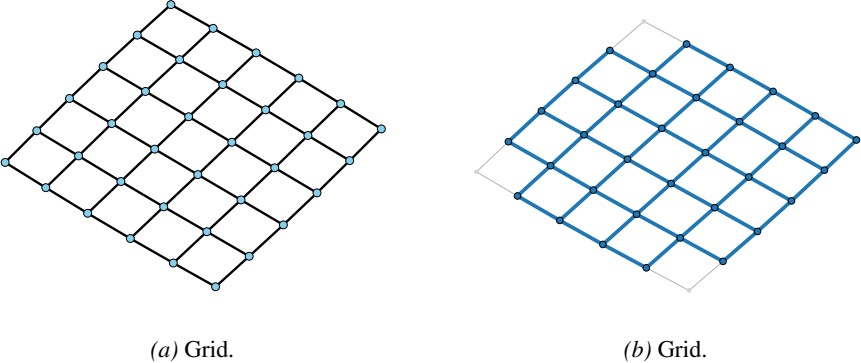

*(a)* Grid.  *(b)* Grid.

*Figure 5.* Visualization of invariant structures on binary tree, grid, path and star graphs.

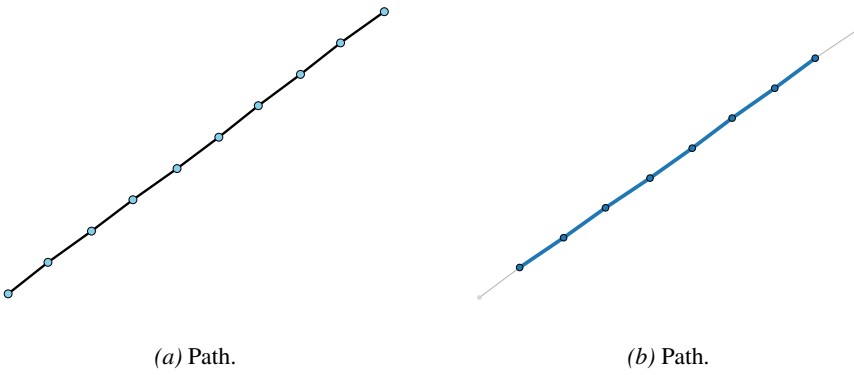

*(a)* Path.  *(b)* Path.

*Figure 6.* Visualization of invariant structures on binary tree, grid, path and star graphs.

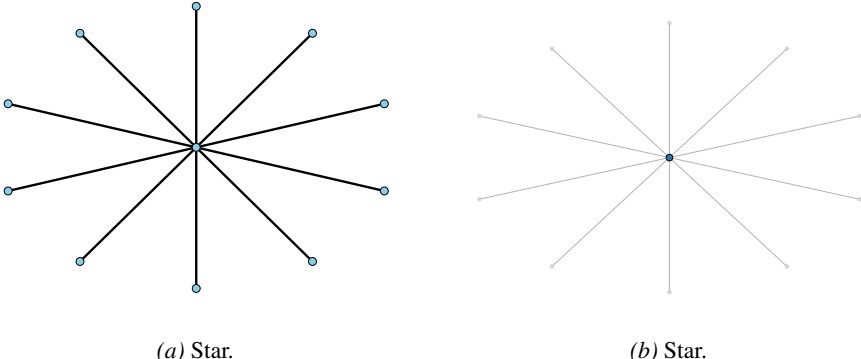

*(a)* Star.               *(b)* Star.

*Figure 7.* Visualization of invariant structures on binary tree, grid, path and star graphs.

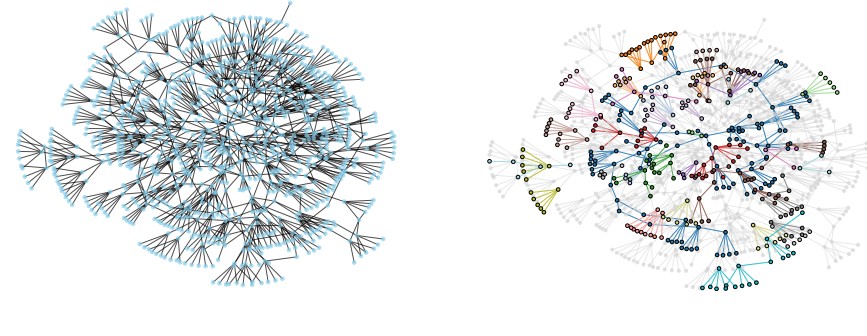

*(a)* Sampled connected subgraph          *(b)* Invariant structures

*Figure 8.* Visualization on Computers.

