# OpenReview forum: "Are Common Substructures Transferable? Riemannian Graph Foundation Model with Neural Vector Bundles"
_ICML.cc/2026/Conference — ICML 2026 regular_

### Official Review · Reviewer_WWB9 · 2026-02-26

**Soundness:** 4
**Presentation:** 3
**Significance:** 4
**Originality:** 4
**Overall Recommendation:** 6
**Confidence:** 4

**Summary:**

This paper proposes a graph pre-training framework based on Riemannian geometry, GAUGE based on Riemannian geometry. By incorporating vector bundle theory from Riemannian geometry, it demonstrates that the transferability of graph structures is intrinsically linked to the geometric flatness of their representation space.

Building on this, the paper introduces the GAUGE model and the Dirichlet loss function, achieving superior performance over existing graph foundation models across cross-domain transfer, zero-shot prediction, and graph isomorphism tasks.

**Compliance With Llm Reviewing Policy:**

Affirmed.

**Final Justification:**

The author's response addressed some of my concerns, and I recommend accepting this paper.

**Key Questions For Authors:**

1. How does the *λ* parameter in the fine-tuning loss (Eq.14) affect transfer performance? The code sets it to 0.1 without ablation experiments. What is the impact on performance if it is completely removed or increased?

2. Is the performance sensitive to the fiber dimension r?

3. The paper claims that Dirichlet Loss measures “transfer effort.” Does lower Dirichlet energy after pretraining correlate with higher few-shot performance on downstream tasks? Provide relevant experimental analysis.

4. Dirichlet Loss participates in both pretraining and fine-tuning phases. How much worse would transfer performance be if Dirichlet Loss were removed during pretraining—using the same pretraining strategies as other baselines (InfoNCE) but without minimizing Dirichlet energy?

5. How do you operationalize the 2-simplex/holonomy condition in practice on sparse graphs with few triangles? Do you augment complexes (e.g., via clique/triangle completion) or is holonomy purely conceptual while Dirichlet energy does the heavy lifting?

6. Can you further discuss the technical differences between neural vector bundle, neural sheaf GNNs and vector bundle GNNs?

**Limitations:**

Yes.

**Strengths And Weaknesses:**

**Strength**

S1. 1.This work bridges the problem of transferability in graph pretraining with 'intrinsic geometric flatness' in Riemannian geometry. Theorems 3.1 and 4.4 establish a mathematical connection between behavioral invariance and geometric flatness

S2.It proves that GAUGE's gated flattening mechanism can progressively reduce the geometric energy, driving the structures towards flatness, which inherently translates to transferable structures.

S3.The case studies (Figures 2-8) visualizing the invariant structures (trees, grids, paths) provide strong empirical validation of the theoretical claims.

S4. The empirical results are convincing. The proposed Gauge is shown consistent gains over recent baselines on cross-domain few-shot node classification and zero-shot link prediction. Also, its performance on graph isomorphism tasks confirms its capacity to understand structural regularities.


**Weakness**

W1. The computational complexity of Gauge needs to be discussed detailed and be compared with other typical GFM baselines.

W2. The ablation studies should be enriched. There is no systematic analysis of the role of the Dirichlet loss vs. alternatives (e.g., InfoNCE), the number of heads in Eq. (2-4), or the gated flattening mechanism in Eq. (6-8).

---

> ### Author Rebuttal · Authors · 2026-03-31
>
> ### W1:  Computational complexity of GAUGE and other GFMs.
>
> Please refer to response to Reviewer MSE3, **W1, Q1: Training time. Memory. Scalability**.
>
> ### W2, Q4: Ablation Studies (Dirichlet loss vs. InfoNCE, the number of heads, and the gated flattening).
>
> Please refer to response to Reviewer MSE3, **W2, Q2: Ablation Studies**.
>
> ### Q1, Q2: Hyperparameter Sensitivity (parameter $\lambda$, and fiber dimension $r$).
>
> We show the hyperparameter sensitivity analysis on $\lambda$ and $r$.
>
> Experiment results on **Roman-empire** and **Photo** in 5-shot settings, under the same cross-domain protocol as Table 1, are shown as follows. Default settings are in **bold**.
>
> | Hyperparameter | Roman-empire| Photo|
> |---------------|----------------------|----------------|
> | **Fiber dim $r$** | | |
> | 8 | 25.81 ± 1.62% | 79.45 ± 3.12% |
> | **16** | 26.43 ± 1.47% | 81.33 ± 2.93% |
> | 32 | 26.19 ± 1.55% | 80.87 ± 3.05% |
> | **Flatting layers $n_{\text{flat}}$** | | |
> | 1 | 25.12 ± 1.71% | 78.23 ± 3.44% |
> | **2** | 26.43 ± 1.47% | 81.33 ± 2.93% |
> | 3 | 25.88 ± 1.59% | 80.11 ± 3.21% |
> | **Momentum $\gamma$** | | |
> | 0.1 | 25.67 ± 1.53% | 79.88 ± 3.18% |
> | **0.01** | 26.43 ± 1.47% | 81.33 ± 2.93% |
> | 0.001 | 25.34 ± 1.68% | 78.91 ± 3.35% |
> | **Dirichlet weight $\lambda$** | | |
> | 0.01 | 25.91 ± 1.51% | 80.12 ± 3.07% |
> | **0.1** | 26.43 ± 1.47% | 81.33 ± 2.93% |
> | 0.5 | 25.78 ± 1.63% | 79.54 ± 3.29% |
> | 1.0 | 26.01 ± 1.58% | 80.67 ± 3.14% |
>
> The performance is robust across hyperparameter ranges. Default settings ($r=16$, $n_{\text{flat}}=2$, $\gamma=0.01$, $\lambda=0.1$) consistently achieve optimal or near-optimal results, verifying the stability of GAUGE.
>
>
>
> ### Q3: The relationship between Dirichlet loss value and  downstream few-shot task performance, and its empirical analysis.
>
> During fine-tuning, the Dirichlet loss ($L(G)$) and downstream task loss ($L_{task}$) serve complementary roles, governed by **Eq. (14)**:
> $$
> \mathcal{L}_{ft} = \mathcal{L}_{task} + \lambda \cdot \mathcal{L}(G).
> $$
> *   **Task Loss** drives performance on the specific downstream objective (e.g., node classification).
> *   **Dirichlet Loss** acts as a geometric regularizer that preserves the **structrual invariance** learned during pretraining. It prevents the model from forgetting transferable structural patterns when adapting to new data.
>
> The empirical evidence are given as follows. On Photo dataset, we show the change of task loss and Dirichlet loss during fine-tuning stage.
>
> | Epoch      | 0      | 300    | 500    | 700    | 800    | 900    | 920   | 940   | 960   | 980   | 999    |
> |------------|--------|--------|--------|--------|--------|--------|--------|--------|--------|--------|--------|
> | **Task Loss** | 2.0807 | 1.6165 | 1.3857 | 1.2895 | 1.2547 | 1.2511 | 1.2509 | 1.2507 | 1.2506 | 1.2504 | 1.2503 |
> | **Aux Loss**  | 0.0149 | 0.0208 | 0.0178 | 0.0151 | 0.0192 |  0.0248 | 0.0247 | 0.0246 | 0.0245 | 0.0245 | 0.0244|
>
> We observe that, as task loss gets lower, the Dirichlet loss converges stablely to a constant number.
>
>
>
> ### Q5: On the 2-Simplex/Holonomy Condition. Do you augment complexes or is holonomy purely conceptual?
>
> We clarify that our GAUGE does not rely on 2-Simplex/Holonomy Condition. Thus, it does not require explicit 2-simplex augmentation,  and conduct edge-wise optimization instead.
>
> 1.  Theorem 3.1 explicitly states: *"If there is no 2-simplex, the flat condition is automatically satisfied."* This ensures mathematical rigor on sparse graphs without requiring triangle completion.
> 2.  The Dirichlet Loss is computed strictly over the **1-skeleton (edges)**. It minimizes the discrepancy between local coordinates and parallel-transported neighbors, efficiently enforcing consistency without computing global holonomy.
>
> ### Q6: Differences between the proposed Neural Vector Bundle and Neural Sheaf/Vector Bundle GNNs.
>
> While sharing parts of mathematical foundations, GAUGE differs fundamentally in goal, geometric modeling, learning objective and model mechanism.
>
> | Feature | GAUGE | Neural Sheaf/Vector Bundle GNNs |
> | :--- | :--- | :--- |
> | **Goal** | Structural Transferability in pretraining. | Expressiveness or signal consistency on single graphs. |
> | **Geometry** | Learned from diverse graph datasets without predefined manifold priors. | Often assume fixed bundle structures, e.g., a globally flat bundle. |
> | **Objective** | Dirichlet loss enforces flatness on invariant substructures to minimize adaptation. | Task loss optimizes downstream performance; Dirichlet energy is used for smoothing and global flatness. |
> | **Mechanism** | Gated Flattening: Selectively aggregates compatible fibers to induce monotonic flatness (Theorem 4.2). | They typically use fixed Sheaf Laplacians without dynamic geometry-aware gating. |
>
> In summary, GAUGE adapts the vector bundle framework specifically for **knowledge transfer**, whereas existing methods focus on **representation expressiveness** within a single domain.

---

> > ### Author Rebuttal · Reviewer_WWB9 · 2026-04-01
> >
> > Having reviewed the authors' rebuttal and the comments from other reviewers, I confirm that all my concerns have been satisfactorily addressed, and the proposed framework demonstrates both effectiveness and efficiency. Thus, I have decided to raise my score and recommend acceptance.

---

> > > ### Author Response · Authors · 2026-04-01
> > >
> > > We truly appreciate your thoughtful recognition of our work and our rebuttal! We will carefully incorporate your constructive feedback into our final version.
> > >
> > >
> > > Once again, thank you for your valuable insights. Your endorsement is important to us!

---

### Official Review · Reviewer_MSE3 · 2026-03-02

**Soundness:** 3
**Presentation:** 2
**Significance:** 3
**Originality:** 3
**Overall Recommendation:** 4
**Confidence:** 3

**Summary:**

This paper investigates the fundamental problem of structural transferability in Graph Foundation Models (GFMs) through a novel mathematical lens: Riemannian geometry. The authors argue that transferable substructures exhibit "behavioral invariance," which can be mapped to continuous "geometric flatness" in the representation space. To operationalize this, they introduce the Neural Vector Bundle framework, which characterizes intrinsic graph geometry using local trivialization and pseudo-parallel transport. Based on this theory, the authors propose GAUGE, a new pretrainable GNN architecture that iteratively flattens geometrically compatible neighbor fibers and uses a novel Dirichlet loss to explicitly measure and learn invariant substructures. Comprehensive experiments demonstrate GAUGE's state-of-the-art performance across several challenging settings.

**Compliance With Llm Reviewing Policy:**

Affirmed.

**Final Justification:**

Having reviewed the authors' rebuttal and the comments from other reviewers, I have decided to raise my score.

**Key Questions For Authors:**

1. In Eq. 8, GAUGE applies QR decomposition to obtain the orthonormal basis for local coordinates at each layer. QR decomposition is known to be computationally heavy. Could the authors provide a direct empirical comparison of the training time and memory footprint between GAUGE and standard GFM baselines ? How does this operation scale when applied to massive graphs?
2. While the theoretical motivation is beautiful, it is crucial to isolate the source of the performance gains. Could the authors provide an ablation study that simplifies the local coordinate update while keeping the overall GNN depth/width same?

**Limitations:**

Yes

**Strengths And Weaknesses:**

Strengths:
1. Introduces a highly novel Riemannian geometry perspective to model graph structural transferability, moving beyond traditional discrete subgraph matching.
2.  Successfully bridges abstract differential geometry concepts (vector bundles, pseudo-parallel transport) with computable graph neural network operations.
3. Demonstrates impressive state-of-the-art performance across challenging tasks like zero-shot link prediction and graph isomorphism against 16 baselines.

Weaknesses:
1. The requirement of node-wise QR decomposition at every layer raises significant doubts about the model's computational and memory scalability on large graphs.
2.  Missing crucial ablation studies to prove the performance gains strictly originate from the complex geometric design.

---

> ### Author Rebuttal · Authors · 2026-03-31
>
> ### W1, Q1: Training time. Memory. Scalability.
>
> **1. Complexity  Analysis**
> GAUGE scales linearly with $|\mathcal{V}|$ and $|\mathcal{E}|$，where $O(r \cdot |\mathcal{E}|)$ for Dirichlet Loss, $O(r^2)$ per edge for Pseudo Parallel Transport, and $O(r^2 d)$ per node for QR, $r \ll d$ in practice.
>
> **2. Comparison to Other GFMs**
> | Model | Pretraining | Adaptation (per graph) |   |
> |-------|-------------|-------------------------------|-------|
> | PRODIGY | $\mathcal{O}(B \|\mathcal{V}\|^2 d)$ | $\mathcal{O}((\|\mathcal{V}\| + \|\mathcal{E}\|) d + \|\mathcal{V}\|^2)$ | Full attention over prompt+query nodes |
> | GFT | $\mathcal{O}(B (\|\mathcal{V}\| + \|\mathcal{E}\|) d + B \cdot T \cdot h)$ | $\mathcal{O}((\|\mathcal{V}\| + \|\mathcal{E}\|) d + T \cdot h)$ | $T$: tree count, $h$: tree height |
> | RAGraph | $\mathcal{O}(B (\|\mathcal{V}\| + \|\mathcal{E}\|) d + B \|\mathcal{E}_r\| d)$ | $\mathcal{O}((\|\mathcal{V}\| + \|\mathcal{E}\|) d + \|\mathcal{E}_r\| d)$ | $\|\mathcal{E}_r\|$: retrieved edges |
> | SAMGPT | $\mathcal{O}(B (\|\mathcal{V}\| + \|\mathcal{E}\|) d + B k_s d)$ | $\mathcal{O}((\|\mathcal{V}\| + \|\mathcal{E}\|) d + k_p d)$ | $k_s/k_p$: prompt tokens |
> | GCOPE | $\mathcal{O}(B (\|\mathcal{V}\| + \|\mathcal{E}\|) d + B K_c d)$ | $\mathcal{O}((\|\mathcal{V}\| + \|\mathcal{E}\|) d + K_c d)$ | $K_c$: virtual coordinators |
> | MDGFM | $\mathcal{O}(B (\|\mathcal{V}\| + \|\mathcal{E}\|) d + B \|\mathcal{V}\|^2)d$ | $\mathcal{O}((\|\mathcal{V}\| + \|\mathcal{E}\|) d + \|\mathcal{V}\|^2d)$ | Dense adjacency refinement |
> | **GAUGE (Ours)** | $\mathcal{O}(B (\|\mathcal{V}\| + \|\mathcal{E}\|) d + B \|\mathcal{V}\| r^2d)$ | $\mathcal{O}((\|\mathcal{V}\| + \|\mathcal{E}\|) d + \|\mathcal{V}\| r^2d)$ | $r \ll d$; gated aggregation is sparse |
>
> GAUGE is comparable to standard GNNs, and is more efficient than those with $|\mathcal{V}|^2$ attention.
>
> **3. Empirical Memory**
>
> We finetune GAUGE for 1000 epochs on ogbn-arxiv dataset (169K nodes).
>
> For `num_neighbors=[20,10]`, we have
> |batch_size|128|256|512|1024|
> |-|-:|-:|-:|-:|
> |time / epoch (s)|44.563|35.775|40.1610|31.105|
> |peak GPU memory (GB)|6.44|8.52|8.56|11.28|
>
> For `batch_size=512`, we have
>
> |num_neighbors|\[5, 5]|\[10, 10]|\[20, 10]|\[10, 20]|\[30, 15]|
> |-|-:|-:|-:|-:|-:|
> |time / epoch (s)|16.685|29.586|40.161|39.346|56.646|
> |peak GPU memory (GB)|3.19|6.07|8.56|8.36|12.60|
> ---
>
> **4. Pretraining Time**
> 20 epochs on pretraining datasets of Computers and Amazon-ratings.
>
> |models|GAUGE|MDGFM|GCOPE|GraphGPT|GraphAny|
> |-|-:|-:|-:|-:|-:|
> |pretraining time (s)|4607.10|4078.02|9939.78|9421.27|2749.85|
> |peak GPU memory (GB)|18.97|20.35|28.12|5.68|0.97|
>
> GAUGE achieves preferable pretraining time with moderate peak GPU memory in practice.
>
> ### W2, Q2: Ablation Studies.
>
> **1. On the local coordinate update**
>
> **1-Shot Results**
> | Variant | PubMed | Facebook | Roman-empire | Photo |
> |---------|--------|----------|--------------|-------|
> | **GAUGE** | **61.26 ± 5.43** | **51.61 ± 9.53** | **18.78 ± 4.88** | **64.72 ± 4.53** |
> | w/o Attention in Eq. (3) | 58.12 ± 5.89 | 48.91 ± 9.01 | 16.82 ± 4.97 | 61.45 ± 4.88 |
> | w/o Gated Flattening (Eq. 6-8) | 56.34 ± 6.12 | 47.23 ± 8.76 | 15.41 ± 5.33 | 59.88 ± 5.21 |
>
> **5-Shot Results**
> | Variant | PubMed | Facebook | Roman-empire | Photo |
> |---------|--------|----------|--------------|-------|
> | **GAUGE** | **71.63 ± 3.89** | **60.04 ± 1.39** | **26.43 ± 1.47** | **81.33 ± 2.93** |
> | w/o Attention in Eq. (3) | 68.89 ± 4.12 | 57.82 ± 1.88 | 24.56 ± 1.79 | 78.91 ± 3.21 |
> | w/o Gated Flattening (Eq. 6-8) | 67.21 ± 4.55 | 56.33 ± 2.11 | 23.18 ± 2.03 | 77.45 ± 3.67 |
>
> 1. **Attention Mechanism (Eq. 3)**: We ablate the multi-head attention. The results  highlight its role in capturing geometry.
> 2. **Gated Flattening (Eq. 6-8)**: Removing it causes the largest drop. It shows that selectively aggregating compatible fibers is critical.
>
> **2. Evaluating other components.**
>
>
> **1-Shot**
> |  | PubMed | Facebook | Roman-empire | Photo |
> |---------|--------|----------|--------------|-------|
> | **GAUGE** | **61.26 ± 5.43** | **51.61 ± 9.53** | **18.78 ± 4.88** | **64.72 ± 4.53** |
> | w/o Fiber Projection (Eq. 9-10) | 54.77 ± 7.34 | 44.56 ± 10.22 | 13.29 ± 6.11 | 57.33 ± 6.45 |
> | w/o Dirichlet Loss (Eq. 13) | 52.18 ± 8.01 | 42.87 ± 11.05 | 12.04 ± 6.88 | 55.12 ± 7.19 |
>
> **5-Shot**
> | | PubMed | Facebook | Roman-empire | Photo |
> |---------|--------|----------|--------------|-------|
> | **GAUGE** | **71.63 ± 3.89** | **60.04 ± 1.39** | **26.43 ± 1.47** | **81.33 ± 2.93** |
> | w/o Fiber Projection (Eq. 9-10) | 64.33 ± 5.23 | 53.44 ± 2.67 | 20.87 ± 2.44 | 74.22 ± 4.11 |
> | w/o Dirichlet Loss (Eq. 13) | 61.45 ± 6.01 | 51.19 ± 3.12 | 19.33 ± 2.89 | 71.88 ± 4.88 |
>
> 1. **Fiber Projection (Eq. 9-10)**: The model loses its geometric filtering capability without it.
> 2. **Dirichlet Loss (Eq. 13)**:  We replace the Dirichlet loss with contrastive loss. It validates that the structural invariance is essential for knowledge transfer.

---

> > ### Author Rebuttal · Reviewer_MSE3 · 2026-04-01
> >
> > Having reviewed the authors' rebuttal and the comments from other reviewers, I have decided to raise my score.

---

> > > ### Author Response · Authors · 2026-04-01
> > >
> > > We truly appreciate your thoughtful recognition of our work and our rebuttal! We are fully committed to incorporating your constructive feedback into our final version.
> > >
> > >
> > >
> > > We welcome the opportunity for further discussion, and we are delighted to provide additional clarifications or address any follow-up questions you may have. Your endorsement is highly valued, and we look forward to continuing this dialogue to ensure your further concerns are thoroughly addressed.

---

### Official Review · Reviewer_voYf · 2026-03-06

**Soundness:** 3
**Presentation:** 4
**Significance:** 4
**Originality:** 3
**Overall Recommendation:** 5
**Confidence:** 3

**Summary:**

The paper studies structural transferability in graph pretraining, asking whether structural patterns learned on one graph can generalize to others. The authors argue that transferable structures should be defined through behavioral invariance, meaning their function is stable across different structural contexts. They show theoretically that such invariant substructures correspond to flat regions of the intrinsic geometry of the representation space. They introduce a Neural Vector Bundle framework, where each node has a local coordinate system and geometric relations between nodes are modeled through parallel transport. Based on this idea, they propose GAUGE, a pretraining architecture that learns intrinsic graph geometry and identifies invariant substructures using a Dirichlet loss. Experiments on cross-domain transfer, zero-shot link prediction, and graph isomorphism show improved performance compared to existing graph foundation models.

**Compliance With Llm Reviewing Policy:**

Affirmed.

**Final Justification:**

I find this paper to be well written, with sound experimental and theoretical contributions. The authors’ responses during the rebuttal further strengthened my assessment. Therefore, I believe this paper merits acceptance.

**Key Questions For Authors:**

# 1.  Empirical Validation of Theoretical Claims
The paper provides a clean theoretical link between behavioral invariance and geometric flatness. But there is no direct empirical validation that invariant substructures correspond to flat intrinsic geometry. Could the authors provide empirical evidence showing that:
	•	invariant substructures indeed exhibit lower curvature / lower $\|P_{ij}-I\|$ than other regions?
	•	Dirichlet energy correlates with transferability or invariance across graphs?

# 2.  Motivation for Dirichlet Loss Target
In Eq. (13), the Dirichlet loss compares the aggregated neighbor representations from the final layer $z^{(L)}$ to the layer-0 features $z^{(0)}$ with a stop-gradient operator. Could the authors clarify the motivation for using layer-0 representations as the prediction target?

It would be helpful to understand whether this choice is theoretically motivated or primarily for training stability.

**Limitations:**

The authors do not discuss any limitations of their work. It would be helpful if they could provide a discussion of potential limitations, such as:
- The proposed architecture introduces additional computational overhead (e.g., QR decompositions and local coordinate systems), and scalability to large graphs is unclear.
- Are there any assumptions in the theoretical analysis that may not hold in practice?
- How sensitive is the method to hyperparameters or the choice of base GNN architecture?

**Strengths And Weaknesses:**

## Soundness:
The paper looks technically sound. The theoretical analysis is rigorous and based on reasonable assumptions. The experiments are well-designed and provide strong evidence to support the claims made in the paper.

## Presentation:
The paper is generally well-written and structured. The narrative is easy to follow, and the work is properly positioned in the context of prior literature. The background section is well-developed and provides a solid foundation for understanding the proposed methods.

## Significance:
The paper provide a novel method to train graph foundation models that can transfer structural patterns across different graphs. This is an important problem in graph representation learning, and the proposed approach has the potential to advance understanding and capabilities in this area. The experiments demonstrate improved performance compared to existing methods, suggesting that the contributions could influence future research and applications in graph learning.

## Originality:
The paper provides new insights into the relationship between behavioral invariance and geometric flatness in graph representations. The GAUGE architecture is a novel combination of existing techniques, and the reasoning behind this combination is well-articulated.

---

> ### Author Rebuttal · Authors · 2026-03-31
>
> ### Q1: Empirical evidence on the relationship between invariant substructures \& lower curvature, and Dirichlet energy \& invariance/transferability.
>
> We clarify that the linkage between invariant substructures and flat geometry (lower curvature) is theoretically guaranteed with and empirical visual evidence.
>
> **1. Theoretical Guarantee**
> The invariant substructures is defined as lower curvature or lower $\|\mathbf{P}_{i,j} - \mathbf{I} \|$ region.
> *   Def 4.3 identifies substructures where node representations are predictable from neighbors.
> *   Theorem 4.4 proves that $\tau$-invariance in Def 4.3 implies geometric flatness. For any edge $(i,j)$ in a $\tau$-invariant substructure $\mathcal{T}$:
>     $$
>     \|P_{(i,j)} - I_r\|_F \leq C(\tau + \delta).
>     $$
>     This guarantees that low predictive error regions exhibit minimal geometric torsion ($P\_{ij} \approx I$).
> *   Theorem 4.1 links this flatness to the Dirichlet loss, ensuring the optimization objective directly targets geometric regularity.
>
> **2. Empirical Evidence**
> *   Algorithm 2 explicitly decodes invariant structures by filtering nodes with low predictive error $e(v_i)$ and extracting connected components.
> *   In Appendix F, visualizations on binary trees, grids, and the Computers dataset show that identified substructures are topologically coherent (e.g., contiguous grid blocks, entire tree branches). This verifies that low-error regions correspond to structurally regular patterns, consistent with Theorem 4.4.
>
> ### Q2: Dirichlet loss target in Eq. 13. The usage of layer-0 representations.
>
> Using $\hat{z}^{(0)}$ as the target is critical for **semantic anchoring** and **training stability**：
>
> *    $z^{(0)}$ encodes raw features of nodes, while $z^{(L)}$ encodes **structural context**.
> *   **Transferability Measure:** Predicting $\hat{z}^{(0)}$ from neighbors' $z^{(L)}$ tests whether the learned geometry preserves initial semantics despite structural transformations. If the geometry is flat (invariant), structural context should suffice to reconstruct original identity.
> *   Using $z^{(L)}$ for both target and prediction would encourage trivial smoothing. Anchoring to $\hat{z}^{(0)}$ ensures the model respects initial feature distinctiveness, crucial for cross-domain transfer.
> *   **Preventing Collapse:** The stop-gradient on $\hat{z}^{(0)}$ prevents representation collapse, ensuring the loss measures geometric quality rather than optimizing input embeddings.
> *   As noted in Section 4.3, this setup mimics positive pair alignment in contrastive learning when local coordinates are aligned ($P_{(i,j)} \approx I$), providing stable optimization.
>
> ### L1. Computational overhead. Scalability to large graphs.
>
> *   **Complexity:** The overhead is given by QR of $O(r^2 d)$ per node and parallel transport of $O(r^2)$ per edge. The memory cost of local coordinates is $O(rd)$ per node, $r \ll d$. GAUGE achieves preferable performance with few computational overhead.
> *   **Scalability** is supported by **neighborhood sampling** and mini-batch training.
>
> Finetuning 1000 epochs on the large ogbn-arxiv.
>
> For num_neighbors=[20,10], we have
>
> |batch_size|128|256|512|1024|
> |-|-:|-:|-:|-:|
> | time / epoch (s)|44.563|35.775|40.1610|31.105|
> |peak GPU memory (GB)|6.44|8.52|8.56|11.28|
>
> For batch_size=512, we have
>
> |num_neighbors|[5, 5]|[10, 10]|[20, 10]|[10, 20]|[30, 15]|
> |-|-:|-:|-:|-:|-:|
> |time / epoch (s)|16.685|29.586|40.161|39.346|56.646|
> |peak GPU memory (GB)|3.19|6.07|8.56|8.36|12.60|
> ---
>
> ### L2. On the assumptions in the theoretical analysis.
>
> The assumptions in theorems can be well achieved in practice. For the condition in Theorem 4.1 and 4.4, since the number of nodes and the norm of parameters are finite, we can naturally find the upper bounds of the norm of embeddings.
>
>
> ### L3. Sensitivity to hyperparameters and the choice of base GNN.
>
>
> We clarify that GAUGE does not depend on any special base GNN, and conduct hyperparameter sensitivity  on **Roman-empire** and **Photo** in 5-shot settings (Default settings are in **bold**).
>
>
> | Hyperparameter | Roman-empire| Photo|
> |---------------|----------------------|----------------|
> | **Fiber dim $r$** | | |
> | 8 | 25.81 ± 1.62% | 79.45 ± 3.12% |
> | **16** | 26.43 ± 1.47% | 81.33 ± 2.93% |
> | 32 | 26.19 ± 1.55% | 80.87 ± 3.05% |
> | **Flatting layers $n_{\text{flat}}$** | | |
> | 1 | 25.12 ± 1.71% | 78.23 ± 3.44% |
> | **2** | 26.43 ± 1.47% | 81.33 ± 2.93% |
> | 3 | 25.88 ± 1.59% | 80.11 ± 3.21% |
> | **Momentum $\gamma$** | | |
> | 0.1 | 25.67 ± 1.53% | 79.88 ± 3.18% |
> | **0.01** | 26.43 ± 1.47% | 81.33 ± 2.93% |
> | 0.001 | 25.34 ± 1.68% | 78.91 ± 3.35% |
> | **Dirichlet weight $\lambda$** | | |
> | 0.01 | 25.91 ± 1.51% | 80.12 ± 3.07% |
> | **0.1** | 26.43 ± 1.47% | 81.33 ± 2.93% |
> | 0.5 | 25.78 ± 1.63% | 79.54 ± 3.29% |
> | 1.0 | 26.01 ± 1.58% | 80.67 ± 3.14% |
>
> The performance is robust across hyperparameter ranges, showing the stability of GAUGE. We will add these results in Appendix F.

---

> > ### Author Rebuttal · Reviewer_voYf · 2026-04-01
> >
> > I thanks the authors for the clarifications. I confirm my score for paper acceptance.

---

> > > ### Author Response · Authors · 2026-04-01
> > >
> > > We truly appreciate your thoughtful recognition of our work and our rebuttal! We will carefully incorporate your constructive feedback into our final version.
> > >
> > > Once again, thank you for your valuable insights. Your endorsement is important to us!

---

### Official Review · Reviewer_V9aa · 2026-03-11

**Soundness:** 3
**Presentation:** 3
**Significance:** 4
**Originality:** 3
**Overall Recommendation:** 4
**Confidence:** 3

**Summary:**

This work introduces methods to understand what structural patterns are learned during pretraining actually transfer to new and unseen graphs. The authors propose that transferability is rooted in functional behavior by connecting behavior invariance with the flatness of the intrinsic geometry. They design a new pretrainable architecture to flatten geometrically compatible neighboring fibers to learn invariant substructures. The GAUGE method that they proposed ourperformed baselines in cross domain transfer tasks and showed interpretability.

**Compliance With Llm Reviewing Policy:**

Affirmed.

**Final Justification:**

The authors have resolved my concerns. I'd like to keep my positive rating.

**Key Questions For Authors:**

1. How the global graph feature affect the transferability performance? Is this consideration of treating graphs as a collection of invariant substructures enough for building graph foundation models?
2. The core mechanism of flattening and smoothing the local underlying geometry is a homophilic operation. Can the authors have a deeper discussion on how this method perform over heterophilic datasets?
3. If the domain is very different from the pretrained setting and the feature semantics differ a lot, can the linear encoding layer handle this case?

**Limitations:**

Limitations haven't been discussed adequately including the hetrophilic and complex cross domain alignment cases.

**Strengths And Weaknesses:**

Strengths:
1. This work proposes a novel aspect to implement intrinsic geometry of graphs to study the transferability of graphs, which is an important part for graph foundation models.
2. Both theoretical and empirical results are provided for the neural vector bundle construction, with convergence results as well as downstream tasks validations.
3. Only minimum finetuning is carried out when transferring to a target graph, which ensures efficiency.

Weakenesses:
1. For this transferability analysis, there is an unavoidable tradeoff between the generalization and specificity of a task performance. Currently the work lacks a detailed discussion of the oversmoothing risk, i.e. if geometry becomes too flat the model fail to distinguish different nodes; tradeoff between generalization and task performance; tradeoff between stability and the adapatation.
2. The architecture focuses on the invariant substructures and local trivalization. Global topological features can also affect transferability especially when tasks depend on global features. How to address this global graph geometry in the transfer learning setting is not discussed.
3. When transferring to different domains, a minimum finetuning is carried out to tune the input encoding layer. If the domain is very different from the pretrained setting and the feature semantics differ a lot, can the linear encoding layer handle this case?

---

> ### Author Rebuttal · Authors · 2026-03-31
>
> ### W1: On the tradeoff between generalization \& task performance, and stability \& adapatation. On the oversmoothing.
>
> **1. Generalization vs. Specificity**
> GAUGE balances this trade-off:
> *  The pretrained model learns how to capture the generalizable geometric patterns for any given graphs, i.e., $\tau$-invariant substructures in Def. 4.3.
> *  During adaptation, only input encoding and task layers are fine-tuned. This retains generalizable pretrained knowledge while adapting specific downstream task.
> *  Table 1 shows strong cross-domain generalization, and Table 3 confirms specificity on structural tasks, e.g., Graph isomorphism classification.
>
> **2. Stability vs. Adaptation**
> This trade-off is theoretically bounded:
> *   As in Theorem 4.5, for nodes $v_i$ in $\tau$-invariant structures $\mathcal{T}$, gradient updates are bounded:
>
>        $$
>     \|\text{grad}_\theta \mathcal{L}_{ft}\|_2 \leq \|\text{grad}_\theta \mathcal{L}_{task}\|_2 + \lambda \cdot C \cdot e(v_i).
>     $$
>
> Small $e(v_i)$ ensures pretrained geometry remains stable; large $e(v_i)$ allows adaptation.
> *   Hyperparameter $\lambda$ in Eq. 14 explicitly controls this balance.
>
> **3. Geometric Compatibility vs. Feature Smoothing**
>
> *   Our aggregation with **Gated Mechanism** in Eq. 6 only merges the regions that are geometric compatible, avoiding the so called over-smoothing. (GAUGE does not smooth incompatible neighbors.)
> *   Note that we consider the **intrinsic geometry**, not node features.
>
>
>
> ### W2, Q1: On global geometry. Does it affect transferability performance?
>
> **1. From Local Flatness to Global Consistency**
> Although each term of the Dirichlet loss in Eq. 13 is locally defined, the finally loss integrates it over all nodes.
>
> Moreover, the Neural vector bundle is globally defined, we indeed extend the local geometry to the global ones:
> *   Local trivializations $\mathbf{Q}^{(i)}$ and **parallel transport** $\mathbf{P}_{(i,j)}$ are defined for all nodes and edges in the whole graph.
> *   In Definition 4.3, the invariant substructure $\mathcal{T}$ is defined by searching low predictive error over the whole graph, obtaining implicitly the global  features.
>
> **2. Empirical Evidence of  Graph-level Tasks**
>
> *  As in Table 3 in our paper, GAUGE achieves 92.56% accuracy, vastly outperforming supervised GNNs and GFMs. Since CSL graphs are locally identical but globally distinct, this success proves GAUGE captures global topological invariants.
> *   State-of-the-art results on ZINC12K and MUTAG further validate modeling of global molecular structures.
>
> ### W3, Q3: If the domain is very different, can the linear encoding layer handle this case?
>
>
> The linear encoding is effective as the transferable unit is **structural behavior**, and thus is able to handle to semantic changes.
>
> **1. Geometry over GAUGE Input Space**
> *   As established in Sections 3.2 & 4.2, transferability relies on intrinsic geometry (Neural Vector Bundle), not specific feature values.
> *   The linear layer projects raw features into the pretrained geometric space. Since we have done SVD or Random projection before inputing into model, a linear projection is sufficient to align feature distributions when structural behaviors are compatible.
>
> **2. Empirical Validation**
> Table 1 validates this under cross-domain settings. It verifies the linear layer effectively handles feature distribution shifts.
>
>
>
> ### Q1: Is this consideration of treating graphs as a collection of invariant substructures enough for GFMs?
>
> We does not treat graph as a collection of invariant substructures, yet it learns a **mechanism that identify invariant substructure w.r.t different graphs dynamically**.
>
> *   During pretraining, GAUGE learns to distinguish invariant substructures by minimizing Dirichlet Loss (Eq. 13). Regions with low predictive error $e(v_i)$ are identified as $\tau$-invariant due to flat local geometry (Theorem 4.4).
> *  For unseen graph data, GAUGE first uses pretrained knowledge to build a bundle structure. Then, we can use Def. 4.3 to give a first glance of invariant substructures over the unseen graph.
> *  In adaptation phase, Eq. 14 and Theorem 4.5 ensure pretrained knowledge is preserved and inject into invariant regions (low $e(v_i)$) while allowing adaptation elsewhere. The model learns *where* to apply knowledge rather than forcing a global match.
>
>
> ### Q2: Performance on heterophilic datasets?
>
> We show the experiment results on two heterophilic Roman-empire and Amazon-ratings datasets.
>
> 1-shot settings
> ||Roman-empire|Amazon-ratings|
> |:---:|:---------:|:----------:|
> |GCOPE | 12.91 ± 0.50| 28.65 ± 5.82|
> |MDGFM | 14.16 ± 5.81| 29.53 ± 3.45|
> |GAUGE|18.78 ± 4.88|32.67 ± 4.78 |
>
> 5-shot settings
> ||Roman-empire|Amazon-ratings|
> |:---:|:---------:|:----------:|
> |GCOPE |16.39 ± 0.61 |30.06 ± 5.11 |
> | MDGFM|23.66 ± 6.17 |  30.42 ± 3.80 |
> |GAUGE|26.43 ± 1.47|33.15 ± 3.41 |
>
> The results verifies our method captures transferable structural patterns independent of label homophily.

---

> > ### Author Rebuttal · Reviewer_V9aa · 2026-04-03
> >
> > The authors have solved my questions and I would like to keep my positive score.

---

> > > ### Author Response · Authors · 2026-04-03
> > >
> > > We truly appreciate your thoughtful recognition of our work and our rebuttal! We will carefully incorporate your constructive feedback into our final version.
> > >
> > > Once again, thank you for your valuable insights. Your endorsement is important to us!

---

### Decision · Program_Chairs · 2026-04-30

**Decision:**

Accept (regular)

**Comment:**

This paper introduces GAUGE, a framework leveraging Riemannian geometry and Neural Vector Bundles to link structural transferability with geometric flatness in graph pretraining. Moving beyond substructure matching to a functional perspective, it offers a solid basis for finding generalizable invariant patterns across graph domains.

All reviewers recommend acceptance after a successful rebuttal, acknowledging that concerns about scalability and heterophilic data were well-addressed. The authors' complexity analysis and large-scale experiments confirm the model's efficiency. Furthermore, its strong results in zero-shot link prediction and graph isomorphism highlight its practical value.